# Implicit Bias of (Stochastic) Gradient Descent for Rank-1 Linear Neural Network

**Bochen Lyu**
DataCanvas Lab, DataCanvas
Beijing, China
bochen.lv@gmail.com

**Zhanxing Zhu**
Changping National Lab & Peking University
Beijing, China
zhanxing.zhu@pku.edu.cn

## Abstract

Studying the implicit bias of gradient descent (GD) and stochastic gradient descent (SGD) is critical to unveil the underlying mechanism of deep learning. Unfortunately, even for standard linear networks in regression setting, a comprehensive characterization of the implicit bias is still an open problem. This paper proposes to investigate a new proxy model of standard linear network, rank-1 linear network, where each weight matrix is parameterized as a rank-1 form. For over-parameterized regression problem, we precisely analyze the implicit bias of GD and SGD—by identifying a "potential" function such that GD converges to its minimizer constrained by zero training error (i.e., interpolation solution), and further characterizing the role of the noise introduced by SGD in perturbing the form of this potential. Our results explicitly connect the depth of the network and the initialization with the implicit bias of GD and SGD. Furthermore, we emphasize a new implicit bias of SGD jointly induced by stochasticity and over-parameterization, which can reduce the dependence of the SGD's solution on the initialization. Our findings regarding the implicit bias are different from that of a recently popular model, the diagonal linear network. We highlight that the induced bias of our rank-1 model is more consistent with standard linear network while the diagonal one is not. This suggests that the proposed rank-1 linear network might be a plausible proxy for standard linear net.

## 1 Introduction

Gradient Descent (GD) and its stochastic variant, Stochastic Gradient Descent (SGD), are probably the most important optimization techniques in deep learning. To unveil the underlying mechanism of modern neural networks, it is highly fundamental to understand the thrilling and mysterious properties of GD and SGD. Some recent works [24, 18, 3, 22] have made significant efforts in this direction by exploring their *implicit bias*: *among all global minimum, i.e., interpolation solutions, what particular ones will GD and SGD prefer without adding an explicit regularization?* This question highlights the crucial roles of these algorithms in the generalization performance of the trained models.

Among the vast number of architectures in deep learning, the first object to investigate is the deep linear network, i.e., without any nonlinear activations,

$$F(x; W) = W_L \cdots W_1 x \qquad (1)$$

where $W_k$'s are weight matrices and $x$ is the input data. The linear network can be obviously treated as an over-parameterized model of standard linear model, $\theta^T = W_L \cdots W_1$. However, the introduction of the over-parameterization brings significant non-convexity, and thus complicates dynamics during learning. Until now, the direct analysis of implicit bias of GD and, particularly, SGD for standard deep linear networks with any depth and initialization on *regression problems* is still an open problem.

37th Conference on Neural Information Processing Systems (NeurIPS 2023).

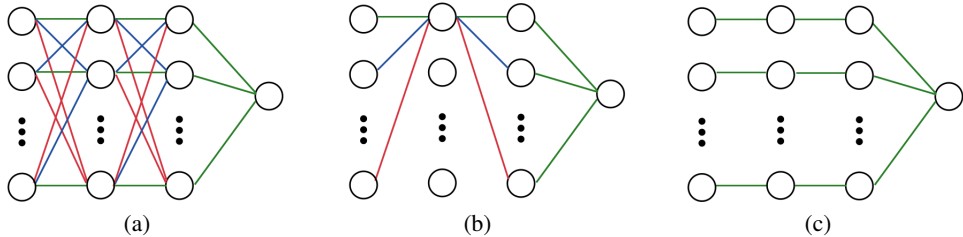

Figure 1: **(a).** Standard linear networks. **(b).** Rank-1 linear networks. One neuron in the middle hidden layer is active and fully connected. **(c).** Diagonal linear networks. Every neuron is active but not fully connected.

In order to study the implicit bias of linear networks in regression settings, a simplified version of standard linear network, diagonal linear network (i.e., each neuron is only connected with a single neuron of the next layer [26, 22]), was proposed as a proxy to underpin the bias of the standard one. With this simplified model, the solution selected by GD solves a constrained norm minimization problem that interpolates between the $\ell_1$ and $\ell_2$ norms up to the initialization scale [26]. For SGD, the solution is closer to that of the sparse regression when compared to GD with the same initialization [22].

However, in Section 3.2, our theoretical analysis shows that the induced implicit bias of diagonal linear networks is not consistent with standard linear network, at least for GD in regression settings. It is then natural to ask: *are there any proxy architectures that could produce similar implicit bias to that of standard linear networks?* If so, we still expect the new proxy model be amenable to tractable analysis and provide insights for investigating the implicit bias of SGD and a wider spectrum of architectures. This is the main goal of our work.

On the other hand, [11] showed that weight matrices of linear networks for linearly separable classification will become low-rank when trained with GD and logistic loss. [16] also observed such low-rank bias for deep matrix factorization. [8] showed that SGD and weight decay jointly induces a low-rank bias in the weight matrices when training a neural network. In fact, for linear networks with a single output trained with GD, all weight matrices will automatically become rank-1 and will maintain this property during training when the initialization is balanced [11].

**Our contributions.** Inspired by this low-rank bias and to avoid the complicated analysis of standard linear networks, we propose a novel model **rank-1 deep linear network** as a plausible proxy of standard linear networks, to reveal the implicit bias of GD and SGD on over-parameterized linear models for regression problems. A depth-$L$ rank-1 deep linear network $f(x; u, v)$ is defined by

$$f(x; u, v) := w_L W_{L-1} \cdots W_1 x = \theta^T x, \ s.t. \ W_k = u_k v_k^T, \ \forall k \in \{1, \ldots, L-1\} \qquad (2)$$

where $u_k \in \mathbb{R}^{d_{k+1}}$ and $v_k \in \mathbb{R}^{d_k}$ are vectors while $u$ and $v$ are denoted as collections of $u_k$ and $v_k$, respectively. See Figure 1 for the difference between three types of linear networks. In this work we will show that the formulation above could offer us the possibility to understand the bias of GD and SGD for standard linear networks through the lens of results on rank-1 linear networks.

With the proposed model, this paper targets on precisely identifying the "potential" function $V(\theta)$ such that GD converges to its minimizer constrained by zero training error (i.e., interpolation solution), and further characterizing the role of the noise induced by adding sampling stochasticity to GD, i.e., SGD, in perturbing the form of $V(\theta)$. For the convenience of theoretical analysis, we focus on the continuous versions of GD and SGD, i.e., we study gradient flow (GF) and stochastic gradient flow (SGF). In particular, this paper establishes the following findings:

- For GF, we show that the solution implicitly minimizes a potential function $V(\theta)$ that depends on the initialization and depth subject to $f(x; u, v)$ achieving zero training error (Theorem 1). The single layer case recovers the standard linear regression results, while a depth larger than one immediately changes the form of $V(\theta)$, which clearly connects the implicit bias of GF with the model architecture. We emphasize that our results explicitly reveal how depth and initialization jointly influence the implicit bias of GF.

More importantly, in Theorem 2, by showing the similarity of the implicit bias of GF for standard and rank-1 linear networks, we conclude that rank-1 linear networks are standard linear networks with special initialization when trained with GF, highlighting that our rank-1 linear network is a plausible proxy of standard linear networks. This offers us the possibility to explore the implicit bias of SGD for standard linear networks by analyzing the rank-1 linear nets. On the other hand, the diagonal linear networks exhibit drastically different implicit bias of GF when compared to standard linear networks, e.g., GF prefers $\ell_2$ minimum norm solution for standard linear networks when the initialization is small while, on the contrary, it prefers such solution when the initialization is large for diagonal linear networks.

- For SGF, we show that the sampling noise brings an extra effect that depends on several hyper-parameters such as the learning rate, batch size and network depth compared to GF. When $L > 1$, this extra effect "alleviates" the influence of the initialization, i.e., the final solution reduces its dependence on the initialization. More intriguingly, when $L = 1$, i.e., without over-parameterization, this effect brought by SGF immediately disappears. Thus *this implicit bias is jointly induced by both model depth and stochasticity from SGD*. To the best of our knowledge, we are the first to show such an implicit bias jointly affected by the two factors through the lens of the rank-1 networks, which is also empirically verified on standard linear and nonlinear networks. Our findings on SGF are summarized in Theorem 3.

**Organization.** In Section 2, we summarize the notations and the setup of our work. Section 3 and Section 4 present our main results where Section 3 focues on the implicit bias of GD while Section 4 is about SGD. Numerical experiments are presented in Section 5 and we conclude this paper in Section 6. Some technical details are deferred to Appendix.

**Related Works**

The study of implicit bias of GD has been pioneered by [24] on linearly separable classification problem and was generalized to GD for deep neural networks and different training strategies [11, 18, 5, 21, 15, 17]. Recently extensive works have made progress in this direction by focusing on the diagonal linear network model. Assuming the existence of a perfect solution in the regression setting, [26] showed that the solution selected by GF interpolates between $\ell_2$ minimum norm solutions and $\ell_1$ ones up to the initialization scale. Besides the full-batch gradient descent, [22] then analyzed SGF for diagonal linear networks and concluded that the sampling noise brought by SGF reduces the effective initialization scale when compared with GF, leading its solution to be closer to a sparse one. Aside from the sampling noise induced by SGF, [9, 23] also investigated the influence of the label noise in the diagonal linear networks setting. For GD and SGD with moderate learning rate, [7] studied their implicit bias for diagonal linear networks and revealed the corresponding influence of the finite learning rate.

This paper considers the rank-1 deep linear network to study the implicit bias of GD and SGD. We briefly introduce differences between the settings considered here and those in previous works. For GD on standard linear networks, [3] did not restrict the initialization for the 2-layer case while [28] required the initialization for standard linear networks to be nearly-zero without restricting the number of layers. [25] considered GD for shallow ReLU nets. In this work, we consider linear networks without requirements either on the scale of the initialization or the number of layers. For GD and SGD on the diagonal linear networks [3, 26, 22], we point out that the results are different with that for standard linear networks. Note that this is not to downgrade diagonal linear network, and instead the point is to show that different architectures could induce different implicit bias. The rank-1 linear networks considered here are close to the standard linear networks since it can be seen as standard linear networks with special initialization. Furthermore, the classification problem [24, 18, 5] and linear regression problem [1] have been comprehensively investigated and we focus on the over-parameterized regression. Finally, [4, 10, 19, 27, 9] focused on the flatness of the loss landscape and model parameters while our analysis is for the overall parametrization $\theta$.

## 2 Preliminaries and Setup

**Notations.** Given a dataset of $n$ samples $\{(x_i, y_i)\}_{i=1}^{n}$, $x_i \in \mathbb{R}^d$ represents a $d$-dimensional data vector with scalar label $y_i \in \mathbb{R}$. We use $X \in \mathbb{R}^{n \times d}$ to denote the data matrix and use $y = (y_1, \ldots, y_n)$

to denote the collection of $y_i$. $\|\cdot\|$ denotes the $\ell_2$-norm and $\|\cdot\|_F$ is the Frobenius norm. $\langle a, b \rangle$ is the inner product. For any vector $u$ and matrix $A$, we use $u(0)$ and $A(0)$ to denote their initialization. We let tr $()$ be the trace operator. For the weight matrix $W_k \in \mathbb{R}^{d_k \times d_{k+1}}$, $W_{k;ij}$ is its $i$-th row $j$-th column element. For a parametric model $f(x; \theta) = \theta^T x$ with parameters $\theta$, we use $\mathcal{L}(\theta) = \frac{1}{n} \sum_i^n \ell_i(\theta)$ to represent the empirical loss where $\ell_i(\theta)$ is the loss function for the sample $(x_i, y_i)$.

**Over-parameterized regression.** We focus on the regression problem where $n < d$ and assume the existence of solutions that perfectly fit the dataset, i.e., there exist interpolating parameters $\theta^*$ such that $\forall i \in \{1, \dots, n\} : \langle x_i, \theta^* \rangle = y_i$. The quadratic loss $\ell_i(\theta)$ is used in our setting. The empirical loss is then $\mathcal{L}(\theta) = \frac{1}{n} \sum_{i=1}^n \ell_i(\theta) = \frac{1}{n} \sum_{i=1}^n (y_i - \langle x_i, \theta \rangle)^2$.

**Rank-1 deep linear networks.** In this paper we consider the rank-1 deep linear network $f(x; u, v)$ (see Eq. (2)). We use $u$ and $v$ to denote the collections of $u_k$'s and $v_k$'s, respectively. For convenience,

$$\rho_k = w_L^T W_{L-1} \cdots W_{k+2} u_{k+1}, \ \rho_{L-1} = 1, \ \text{and} \ \rho_{-k} = v_k^T W_{k-1} \cdots W_2 u_1, \ \rho_{-1} = 1 \quad (3)$$

such that the network can be written as $f(x; u, v) = \rho_k v_{k+1}^T u_k \rho_{-k} v_1^T x$ for $k \in \{1, \dots, L-1\}$ when $L \geq 2$. Through this paper, we treat the depth $L$ as a hyper-parameter of the network and try to precisely characterize its role in the implicit bias.

**Definition 1** (Balanced initialization for rank-1 linear networks). *Given an $L$-layer rank-1 linear network Eq. (2), for any $k \in \{1, \dots, L-1\}$, the balanced initialization means that*

$$\frac{\langle v_{k+1}(0), u_k(0) \rangle^2}{\|v_{k+1}(0)\|^2 \|u_k(0)\|^2} = 1, \ \|v_{k+1}(0)\| = \|u_k(0)\| = \|v_1(0)\|.$$

We add more discussions on the initialization in Appendix C.1.

## 3 Equivalence Between Implicit Bias of GD for Standard and Rank-1 Nets

### 3.1 Implicit bias of GF for rank-1 linear networks

In this section, we characterize the implicit bias of the continuous version of GD for the rank-1 deep linear networks $f(x; u, v)$ Eq. (2). Note that the model parameters are updated according to the gradient flow

$$\frac{du_k}{dt} = -\nabla_{u_k} \mathcal{L}(\theta), \ \frac{dv_{k+1}}{dt} = -\nabla_{v_{k+1}} \mathcal{L}(\theta) \quad (4)$$

for $k \in \{1, \dots, L-1\}$ and $dv_1/dt = -\nabla_{v_1} \mathcal{L}(\theta)$, which clarifies that $u_k$ and $v_k$, rather than $W_k$, are the model parameters.

Previous work [26] showed that the solution selected by GF interpolates between the $\ell_1$ minimum norm solution and $\ell_2$ one depending on the initialization scale for diagonal linear networks. In this section we examine whether this is the case for the rank-1 deep linear network with depth $L$. For convenience, we define

$$\Omega_L = \frac{2L}{2L-1}, \ \lambda_L = \frac{2(L-1)}{2L-1} \quad (5)$$

where $L$ is the number of layers. Recall that $\theta(0)$ represents the initialization of $\theta$, we now state the main theorem regarding the implicit bias of GF for rank-1 linear networks below:

**Theorem 1** (Implicit bias of GF for rank-1 linear networks). *For a rank-1 linear network, if the initialization is balanced across layers (Definition 1) and if the gradient flow solution $\theta(\infty)$ satisfies that $X\theta(\infty) = y$, then gradient flow converges to a minimizer of the potential function $V(\theta)$:*

$$\theta(\infty) = \arg\min_\theta V(\theta), \quad s.t. \ X\theta = y,$$

*where*

$$V(\theta) = \frac{1}{\Omega_L} \|\theta\|^{\Omega_L} - \theta^T \frac{\theta(0)}{\|\theta(0)\|^{\lambda_L}}. \quad (6)$$

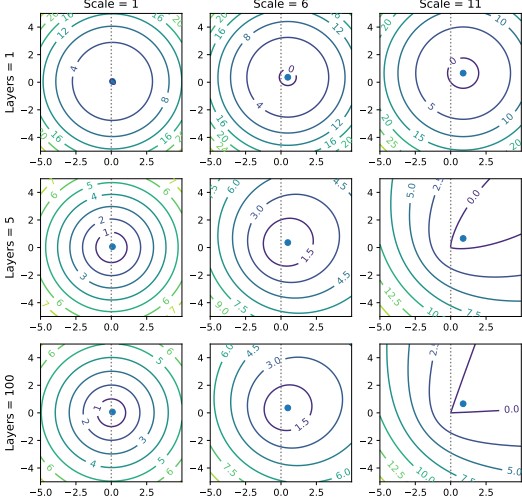

Figure 2: The plot of $V(\theta)$ for different numbers of layers and initialization scales in a two-dimensional parameter space. The initialization in the first column is [0.08, 0.06] and is multiplied by different scale factors for the other two columns. In the first column where the initialization is nearly zero, GD has similar implicit bias for rank-1 linear networks of different layers. When we increase the initialization scale, the shape of the contour for linear regression potential function does not change, while for linear networks as shown in the last column, the contour of the potential function gradually presents a sharp angle as we increase the number of layers.

While previous works focused on 2-layer standard linear networks [3] or multiple-layer linear networks with nearly zero initialization [28], i.e., $\|\theta(0)\| \approx 0$, Theorem 1 builds the potential function $V(\theta)$ that *depends on both the initialization $\theta(0)$ and the depth of the network $L$ explicitly*. We also plot the contours of $V(\theta)$ for different numbers of layers and scales of initialization in Fig. 2 in a 2-dimensional space. Theorem 1 clearly reveals how over-parameterization and the initialization of $\theta$ guide GD to select different solutions.

**Effects of depth.** When $L = 1$, $V(\theta)$ becomes $\frac{1}{2}\|\theta\|^2 - \langle\theta, \theta(0)\rangle$ which is the same as the potential of the least square $\|\theta - \theta(0)\|^2$ except for a constant term. When $L > 1$, $V(\theta)$ is no longer an Euclidean distance. The form of $V(\theta)$ depends on the depth and GD will favor different interpolating solutions. A particular interesting case is when $L \to \infty$ where we have $\Omega_L \to 1$ and $\lambda_L \to 1$. As a result, $V(\theta)$ becomes $V(\theta) \to \|\theta\| - \langle\theta, \theta(0)/\|\theta(0)\|\rangle$, which reflects the difference between the norm of $\theta$ and the norm of its projection on the direction of the initialization $\theta(0)$. Therefore, the direction of the initialization $\theta(0)$ matters for the potential function $V(\theta)$ and the final solution $\theta(\infty)$ while they are rather less sensitive to the scale of $\theta(0)$. This may serve as a benefit of the over-parameterization in the sense that it makes the network more stable to different scales of the initialization. As a comparison, both direction and scale of $\theta(0)$ are crucial for the least square potential function $\|\theta - \theta(0)\|^2$.

**Effects of initialization.** When $L$ is finite, to inspect the effects of the initialization, we can rewrite $V(\theta) = \|\theta\|^{\Omega_L}/\Omega_L - \|\theta(0)\|^{1/(2L-1)} \langle\theta, \theta(0)/\|\theta(0)\|\rangle$. The second term will be more important when the initialization scale $\|\theta(0)\|$ is getting larger thus the initialization affects the implicit bias of GD more significantly. On the other hand, as $\|\theta(0)\| \to 0$, the second term vanishes. Thus we have the following corollary:

**Corollary 1.1.** *Under conditions of Theorem 1, if we further assume that the initialization is infinitesimal $\|\theta(0)\| \to 0$ and the depth is finite, then the GF solution $\theta(\infty)$ is an $\ell_2$-norm minimization solution:*

$$\theta(\infty) = \min_\theta \|\theta\| \quad s.t. \ X\theta = y.$$

We now summarize the effects of initialization on the training regime as follows. **(i).** According to [26], for any $D$-homogeneous model ($D$ is a positive integer), the lazy regime (or NTK regime) is reached for large initialization. Since the rank-1 linear network is a homogeneous model, the NTK regime is reached for large initialization and the implicit bias is given by the RKHS norm predictor accordingly; **(ii).** For vanishing initialization, according to Corollary 1.1, an $\ell_2$-norm minimization predictor which can not be captured by the NTK kernel is returned. Therefore, as the initialization becomes smaller, we "escape" from the lazy regime and falls into the "Anti-NTK" regime defined

Figure 3: GD maintains the shape of rank-1 initialization for the weight matrices (the upper panel), but destroys that of the diagonal initialization for standard linear networks (the lower panel), as formally described by Proposition 1.

by [3]. Furthermore, based on Corollary 1.1, it is now worth to mention that, assuming vanishing initialization, linear regression (i.e., 1-layer linear networks), 2-layer standard linear networks [3] and rank-1 linear networks with finite depth share a similar implicit bias if trained with GD. And the first column of Fig. 2 shows that rank-1 linear networks with different number of layers exhibit potential contours with a similar shape.

## 3.2 Comparison between different architectures

The aforementioned phenomenon that GD will return an $\ell_2$-norm minimization solution for both 2-layer standard linear networks and rank-1 linear networks with small initialization drives us to further investigate the comparison and similarities between different network architectures. For diagonal linear networks, GD prefers $\ell_1$-norm minimization solution with nearly-zero initialization, which is drastically different from the case for rank-1 and standard linear networks. On the contrary, it prefers $\ell_2$-norm minimization solution when the initialization is sufficiently large [26]. This inconsistency begs us to ask the question *will standard linear networks and rank-1 linear networks share a similar implicit bias of GD when the initialization is large?* To answer this question, in the following, we first analyze the implicit bias of GD for standard linear networks $f(x; W) = w_L^T W_{L-1} \cdots W_1 x$ with any depth $L$, which corresponds to the parameterization $\theta^T = w_L^T W_{L-1} \cdots W_1$, with balanced initialization (Definition 2).

**Theorem 2** (Implicit bias of GF for standard linear networks with any initialization). *For an $L$-layer standard linear network, if the initialization is balanced across layers, i.e., $W_{k+1}^T(0)W_{k+1}(0) = W_k(0)W_k^T(0)$ for all layers, and if the gradient flow solution $\theta(\infty)$ satisfies that $X\theta(\infty) = y$, then gradient flow converges to a minimizer of the potential function $V(\theta)$:*

$$\theta(\infty) = \arg\min_\theta V_{std}(\theta), \quad s.t. \ X\theta = y,$$

*where*

$$V_{std}(\theta) = \frac{L}{L+1}\|\theta\|^{\frac{L+1}{L}} - \theta^T \frac{\theta(0)}{\|\theta(0)\|^{\frac{L-1}{L}}}, \tag{7}$$

Although we state before that the parameters of a rank-1 deep linear network are $u_k$'s and $v_k$'s rather than $W_k$'s, the form of the potential for rank-1 linear networks Eq. (6) is very similar to that of standard linear networks Eq. (7): $V(\theta)$ for an $L$-layer rank-1 linear network is the same as $V_{std}(\theta)$ for a $(2L-1)$-layer standard linear network. In the following, we explain the reason behind this fact and conclude that rank-1 linear network can be seen as a qualified proxy for the standard linear network when studying the implicit bias of GD, and potentially SGD.

**Rank-1 linear networks are standard linear networks with special initialization.** We first present a useful proposition regarding a special kind of initialization for standard linear networks.

**Proposition 1** (Effects of diagonal and rank-1 initialization for standard linear networks). *Given an $L$-layer standard linear network $f(x; W) = w_L^T W_{L-1} \cdots W_1 x$ where $W_k \in \mathbb{R}^{d_{k+1} \times d_k}$ for $k \in \{1, \ldots, L\}$, if the weights are initialized such that only one column of $W_k$ is non-zero when $k$ is even and only one row of $W_k$ is non-zero otherwise, i.e., for an integer $p$*

$$\forall k = 2p+1 \in \{1, \ldots, L\} : W_{k;ij}(0) = 0 \ \text{if } i \neq c_k, \ \text{where } c_k \in \{1, \ldots d_{k+1}\}$$
$$\forall k = 2p \in \{1, \ldots, L\} : W_{k;ij}(0) = 0 \ \text{if } j \neq c_k,$$

*then for any $t > 0$:*

$$\forall k = 2p + 1 \in \{1, \ldots, L\} : W_{k;ij}(t) = 0 \text{ if } i \neq c_k;$$
$$\forall k = 2p \in \{1, \ldots, L\} : W_{k;ij}(t) = 0 \text{ if } j \neq c_k.$$

*Furthermore, if the weights are initialized as the diagonal shape, i.e., $\forall k \in \{1, \ldots, L - 1\}, W_{k;ij}(0) = 0$ if $i \neq j$, then GD does not maintain the diagonal shape of weight matrices.*

**Remark.** A $(2L - 1)$-layer standard linear network initialized as in Proposition 1 with $c_k = 1$ for all layers has the same formulation as our $L$-layer rank-1 architecture. According to Proposition 1, the special structure of the initialization of such standard linear network will be maintained if we run GD, which suggests that it will always have the same formulation as our $L$-layer rank-1 linear network, see the upper panel of Fig. 3. Therefore, an $L$-layer rank-1 linear network can be seen as a $(2L - 1)$-layer standard one with special initialization, and they exhibit similar implicit bias of GD. On the other hand, if the standard linear network is initialized as the diagonal shape as in the diagonal net, GD will not maintain this property (the lower panel of Fig. 3), i.e., the diagonal structure cannot be seen as a standard one. In this sense, the diagonal network exhibits special implicit bias particularly due to its structure, while the rank-1 linear network is a more qualified proxy of standard linear networks.

## 4 Implicit bias of SGD for Rank-1 Linear Networks

Recently, [23, 22] developed the characterization of the implicit bias of GD with label noise and SGD in diagonal linear networks. However, as mentioned earlier, its diagonal structure is special in the sense that the corresponding results can not be generalized to the case for standard linear networks. On the contrary, our results in Section 3 reveal that rank-1 linear networks can be seen as standard linear networks with special initialization. In this section, to take a step forward towards understanding the implicit bias of SGD for standard linear networks, we explore the continuous part of SGD, stochastic gradient flow (SGF), for the rank-1 linear networks in the over-parameterized regression setting. We begin with the introduction of the definition of SGD and our modelling techniques.

**SGD.** Unlike the full-batch GD where the parameters are updated according to Eq. (4), the SGD dynamics is

$$u_k(t + 1) = u_k(t) - \frac{\eta}{b} \sum_{i \in \mathcal{B}_t} \nabla_{u_k} \ell_i(\theta), \quad v_{k+1}(t + 1) = v_{k+1}(t) - \frac{\eta}{b} \sum_{i \in \mathcal{B}_t} \nabla_{v_{k+1}} \ell_i(\theta) \qquad (8)$$

for $k \in \{1, \ldots, L - 1\}$ where $t$ denotes the iteration step, $\eta$ is the learning rate, $b$ is the batch-size, and $\mathcal{B}_t$ consists of $b$ points randomly sampled from the uniform distribution $\mathcal{U}[1, n]$.

**Continuous Modelling of SGD.** The continuous modelling techniques for SGD have been widely applied in recent works [1, 9, 23, 22] to study the dynamics of SGD. In our setting, recalling the definition of $\rho_k$ and $\rho_{-k}$ in Eq. (3), the continuous counterpart of SGD, SGF, is given by the following set of stochastic differential equations (SDE):

$$du_k = -\frac{2}{n} v_1^T X^T r \rho_k \rho_{-k} v_{k+1} dt + 2\sqrt{\frac{\eta \mathcal{L}}{nb}} (\rho_k \rho_{-k}) v_{k+1} v_1^T X^T d\mathcal{W}_t \qquad (9)$$

$$dv_{k+1} = -\frac{2}{n} v_1^T X^T r \rho_k \rho_{-k} u_k dt + 2\sqrt{\frac{\eta \mathcal{L}}{nb}} (\rho_k \rho_{-k}) u_k v_1^T X^T d\mathcal{W}_t \qquad (10)$$

where $r = (f(x_1; u, v) - y_1, \ldots, f(x_n; u, v) - y_n)^T \in \mathbb{R}^n$ is the residual, and $\mathcal{W}_t$ is a standard Brownian motion in $\mathbb{R}^n$. For the parameterization of $\theta$ (Eq. (2)), we now aim to characterize the implicit bias of SGD by showing the existence of a function $V(\theta)$ such that the solution of the stochastic dynamics converges to its minimizer under the constraint of zero-training loss as follows.

**Theorem 3** (Implicit bias of SGF for rank-1 linear networks)**.** *For the rank-1 linear network Eq. (2) that is trained with SGF (Eq. (9) and (10)) in the over-parameterization regression, if the initialization is balanced across layers, then the dynamics of $\theta$ gives us*

$$d\nabla_\theta V^{\mathcal{S}}(\theta, t) = -\nabla_\theta \mathcal{L} dt + 2\sqrt{\frac{\eta |\xi| \mathcal{L}}{nb}} X^T d\mathcal{W}_t \qquad (11)$$

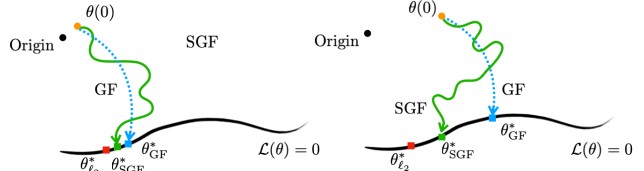 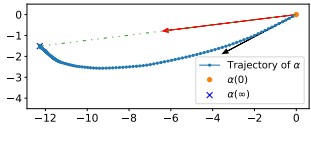

(a) The implicit bias jointly induced by stochasticity and architectures "alleviates" the effects of initialization

(b) The trajectory of $\alpha$ starting from the origin converging to $\alpha(\infty)$

Figure 4: **(a)** The blue trajectory is for gradient flow (GF) that converges to $\theta_{\mathrm{GF}}^*$ (blue square). The green trajectory that converges to $\theta_{\mathrm{SGF}}^*$ (green square) is for stochastic gradient flow (SGF). $\theta(0)$ denotes the initialization of $\theta$ and $\theta_{\ell_2}^*$ (red square) is the $\ell_2$-norm minimum solution. Black dot marks the origin. Compared to $\theta_{\mathrm{GF}}^*$, $\theta_{\mathrm{SGF}}^*$ is closer to $\theta_{\ell_2}^*$ since its dependence on the initialization is reduced. Note that when $\theta(0)$ is nearly zero (the left figure), both $\theta_{\mathrm{GF}}^*$ and $\theta_{\mathrm{SGF}}^*$ are close to $\theta_{\ell_2}^*$ (Corollary 1.1). **(b)** The red arrow denotes the direction of $\alpha(\infty)$ while the black one denotes that of $\theta(0)$, which, as shown in the figure, is particularly important to the direction of $\alpha(\infty)$. See more experiments in Appendix A.3.

*where*

$$V^{\mathcal{S}}(\theta, t) = \frac{1}{\Omega_L}\|\theta\|^{\Omega_L} + \frac{\theta^T\theta(0)}{\|\theta(0)\|^{\lambda_L}} - \frac{2\lambda_L\eta\theta^T}{nb}\int_0^t \frac{\mathcal{L}\left(\theta(s)\right)\operatorname{tr}\left(P_\perp(\theta(s))X^TX\right)}{\|\theta(s)\|^{2-\lambda_L}}\theta(s)ds \quad (12)$$

*with $P_\perp(\theta) = I - \theta\theta^T/\|\theta\|^2$ being the orthogonal projection operator of $\theta$. Furthermore, if the SGF solution $\theta(\infty)$ satisfies that $X\theta = y$, then the model parameter $\theta$ converges to a minimizer of $V^{\mathcal{S}}(\theta, \infty)$ as $t \to \infty$:*

$$\theta(\infty) = \arg\min_\theta V^{\mathcal{S}}(\theta, \infty), \quad s.t. \ X\theta = y.$$

One can immediately notice that the R.H.S of Eq. (11) includes a noise term, which can be interpreted as that the model parameter $\theta$ follows a mirror flow with a noise term. Furthermore, Eq. (12), which characterizes the optimization geometry, depends on time explicitly. This partly reflects its stochastic nature—the optimization trajectory is not deterministic. Furthermore, as in [7], it is also possible to generalize the current vanishing learning rate results to the moderate learning rate analysis by deriving a stochastic mirror descent recursion with time varying potentials.

**Implicit bias jointly induced by stochasticity and architectures.** The sampling noise of SGD introduces an extra term (the last term of Eq. (12)) that depends on several parameters such as the learning rate and the data matrix $X$ when compared with that of GD (Eq. (6)). Interestingly, this extra term only exists when the model is *simultaneously* over-parameterized and trained with the existence of the sampling noise. If the model is not over-parameterized, i.e., standard linear regression when $L = 1$, then $\lambda_L = 0$ and this term disappears. On the other hand, if there is no sampling noise this term also vanishes. Indeed, the $L = 1$ case has been widely studied by recent works [1], and it turns out that GD and SGD share similar implicit bias when there is no over-parameterization. Theorem 3 confirms this phenomenon and, more importantly and intriguingly, reveals a new connection between the architecture-induced over-parameterization and the implicit bias of the optimization algorithms.

**"Alleviating" the effects of initialization.** When $L \to \infty$, the last term of Eq. (12) brought by the sampling noise can be seen as *"alleviating" the influence of the initialization $\theta(0)$ such that the training dynamics reduces its dependence on the initialization* (Fig. 4(a)). Due to the difficulty of explicitly solving the stochastic integral in Eq. (12), we give here a qualitative interpretation of this alleviating effect. As $L \to \infty$, i.e., infinitely deep linear network, we have $\lambda_L \to 1$, therefore the integral in Eq. (12) becomes $-\frac{2\eta}{nb}\int_0^\infty \mathcal{L}(\theta)\operatorname{tr}\left(P_\perp(\theta)X^TX\right)\frac{\theta}{\|\theta\|}ds$. In a $d$-dimensional space, we let $\alpha(t) := \int_0^t \mathcal{L}(\theta)\operatorname{tr}\left(P_\perp(\theta)X^TX\right)\theta/\|\theta\|ds$ be the position of a particle $A$ that starts from the origin. Then the integral of Eq. (12) amounts to the final position $\alpha(\infty)$ of $A$, which moves

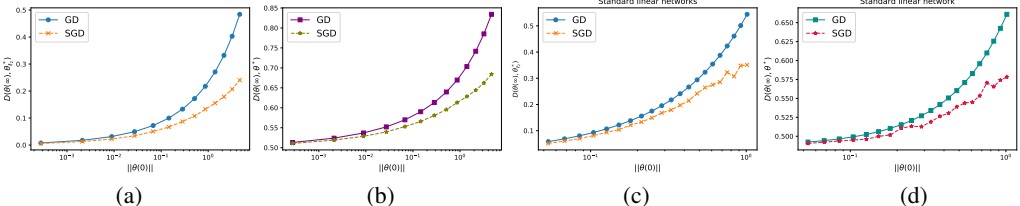

| (a) | (b) | (c) | (d) |

Figure 5: For different $\|\theta(0)\|$: **(a)** $D(\theta(\infty), \theta^*_{\ell_2})$ for rank-1 linear networks. **(b)** $D(\theta(\infty), \theta^*)$ for rank-1 linear networks. **(c)** $D(\theta(\infty), \theta^*_{\ell_2})$ for standard linear networks. **(d)** $D(\theta(\infty), \theta^*)$ for standard linear networks.

along the direction of $\theta(t)/\|\theta(t)\|$ with speed proportional to $\mathcal{L}(\theta(t)) \operatorname{tr}\left(P_\perp(\theta(t)) X^T X\right)$. Since $\|X\|_F^2 - \|X\|_2^2 \leq \operatorname{tr}\left(P_\perp(\theta(t)) X^T X\right) \leq \|X\|_F^2$ where $\|X\|_F^2$ is finite, it is highly likely that the direction of $\alpha(\infty)$ heavily depends on $\theta(0)$. This is because the velocity magnitude, which is proportional to the value of empirical loss $\mathcal{L}(\theta(t))$, is large when $t = 0$ and quickly shrinks along its trajectory. As a result, in Eq. (12), the effect coming from the initialization $\theta^T \theta(0)/\|\theta(0)\|$ is alleviated by the extra term $-\theta^T \alpha(\infty)$ induced by the sampling noise and over-parameterization. In this sense, $V^{\mathcal{S}}(\theta, \infty)$ is closer to a simple $\ell_2$-norm $\|\theta\|$ when compared with GD, i.e., the SGD solution is more likely to be an $\ell_2$ minimum norm solution when compared to GD. We also present the trajectory of $\alpha$ in Fig. 4(b). The trajectory is obtained by first training a rank-1 linear network using SGD for 5000 iterations and computing $\alpha(t)$ at every step followed by a projection of the trajectory $\{\alpha(t)\}_{t=1}^{5000}$ into a 2-dimensional space according to the method described in [13]. It can be seen that the direction of the final position $\alpha(\infty)$ is close to that of $\theta(0)$ ($\langle\alpha(\infty), \theta(0)\rangle /(\|\alpha(\infty)\|\|\theta(0)\|) = 0.9314$). We also conduct additional experiments in Appendix A.3 to further verify our finding.

**Comparison with diagonal linear networks.** In [22], the authors showed that the extra effects of the sampling noise of SGD is equivalent to multiplying a shrinking coefficient which depends on the training dynamics to the initialization scale. This effect leads the solution of SGD to be closer to that of sparse regression when compared with GD. Our Theorem 3 does not show such bias that the solution interpolates between the $\ell_2$-norm and $\ell_1$-norm minimization solution. Therefore the conclusion of implicit bias of SGD for diagonal linear networks can not be directly applied to rank-1 linear networks and standard linear networks. This reveals an important finding that the implicit bias of GD or SGD is strongly tied with network architecture. On the other hand, similar to the case of diagonal linear networks, Theorem 3 also reveals an initialization cancellation effect induced by the SGD sampling noise. It is interesting for future work to explore whether this effect can be generalized to other architectures and can be seen as a special benefit brought by SGD.

## 5 Numerical Experiments

In this section, we consider the over-parameterized regression problem with rank-1 linear, standard linear, and non-linear networks for different initialization scales to verify our theoretical claims. We define the distance $D(\theta_1, \theta_2) = \|\theta_1 - \theta_2\|^2/\|\theta_2\|^2$ to measure the relative difference between $\theta_1$ and $\theta_2$. For a linear network and the parameterization $\theta^T = w_L^T W_{L-1} \cdots W_1$, we use $\theta(\infty)$ to denote the solution returned by the optimization algorithms (GD or SGD) and $\theta(0)$ to denote the corresponding initialization. $\theta^*$ is the ground truth solution, and $\theta^*_{\ell_2}$ is the $\ell_2$ minimum norm solution (Corollary 1.1). Details of the experiments and more numerical experiments are deferred to Appendix A.1 and we focus on the results here.

**Rank-1 linear networks.** As shown in Fig. 5(a), for all different initialization scales $\|\theta(0)\|$, $D(\theta(\infty), \theta^*_{\ell_2})$ is smaller if $\theta(\infty)$ is returned by SGD, i.e., the SGD solution is closer to the $\ell_2$ minimum norm solution $\theta^*_{\ell_2}$ when compared to the GD solution. Similarly, Fig. 5(b) shows a benefit of such alleviating dependence of initialization bias of SGD: its solution is closer to the ground truth solution $\theta^*$ when compared to the GD solution for different $\|\theta(0)\|$. Furthermore, in Fig. 6(a), we show $D(\theta(t), \theta^*_{\ell_2})$ along training, which further reveals that the final solution returned by SGD is closer to $\theta^*_{\ell_2}$. Note that when the initialization scales are small, both GD solution and SGD solution

are close to $\theta_{\ell_2^*}$. These experiments well support our theoretical findings. Besides, we present additional experiments to compare diagonal and rank-1 linear networks in Appendix A.2.

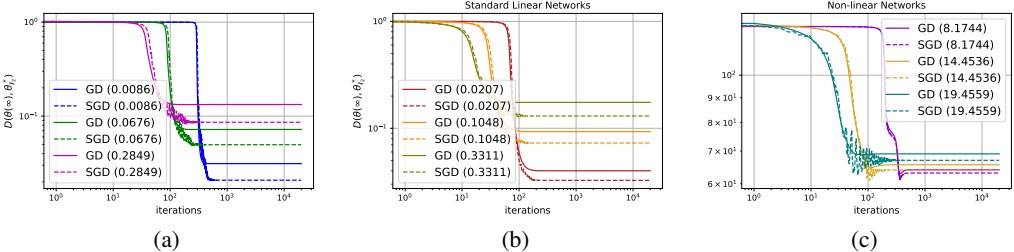

Figure 6: Numbers in the bracket denote the scale of the initialization. **(a).** $D(\theta(t), \theta_{\ell_2}^*)$ along training for rank-1 linear networks. **(b).** $D(\theta(t), \theta_{\ell_2}^*)$ along training for standard linear networks. **(c).** Test error along training for non-linear networks.

**Standard linear networks and non-linear networks.** Proposition 1 states similarity between rank-1 and standard linear nets, therefore they have similar results. We conduct numerical experiments for standard linear networks as in the case of rank-1 linear networks and show in Fig. 5(c) and 5(d) that $D(\theta(\infty), \theta_{\ell_2}^*)$ and $D(\theta(\infty), \theta^*)$ are also smaller if we run SGD, which supports the generalization of conclusion of rank-1 linear network to standard linear networks. $D(\theta(t), \theta_{\ell_2}^*)$ along training plotted in Fig. 6(b) further supports this phenomenon. For non-linear networks (Fig. 6(c)), we report the test error on a newly sampled test set for both GD and SGD along training, where SGD solutions have smaller test error when compared to GD solutions.

## 6 Discussion & Conclusion

Our work proposes the rank-1 linear network that is a plausible proxy of standard linear networks. We analyze the implicit bias of GD and SGD for this new net and find that it approximates standard linear networks better than diagonal linear networks. We further reveal the joint role of over-parameterization and stochasticity in characterizing the implicit bias of SGD. Similar to the diagonal linear networks, our results also reveal an initialization alleviating effect of SGD sampling noise, suggesting a future direction that investigates whether such effect is general across different architectures. See Appendix B for more discussions.

**Limitation.** We do not generalize the analysis to non-linear networks due to its lack of the form of $\theta$ as in our current approach. Furthermore, an exact characterization of the stochastic integral is absent.

## Acknowledgments

B. Lyu thanks Di Wang for useful discussion. Z. Zhu is supported by Beijing Nova Program (No. 202072) from Beijing Municipal Science Technology Commission.

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

# Appendix

**Organization of Appendix.** In Appendix A we present details of the numerical experiments and additional numerical experiments. We give more discussions of our work in Appendix B. Appendix C provides missing technical details of Section 3 while Appendix D provides those of Section 4.

## A Details of Numerical Experiments and Additional Experiments

In A.1, we present the details of the numerical experiments. We compare the rank-1 linear networks with the diagonal linear networks empirically in A.2. Finally, in A.3, we conduct additional experiments to further verify the "alleviting" effect of the SGD sampling noise mentioned in Theorem 3. In A.4, we conduct experiments when the balanced initialization condition is not satisfied.

**Data.** We conduct over-parameterized regression with different linear networks. For the dataset $\{(x_i, y_i)\}_{i=1}^n$ where $x_i \in \mathbb{R}^d$ and $y_i \in \mathbb{R}$, we set $n = 40, d = 100$ and $x_i \sim \mathcal{N}(0, I)$. For $i \in \{1, \ldots, n\}$, $y_i$ is generated by $y_i = \theta^{*T} x_i$ where $\theta^* \in \mathbb{R}^d$, i.e., $\theta^*$ is the ground truth solution. We let 20 components of $\theta^*$ be informative.

### A.1 Details of Numerical Experiments in Section 5

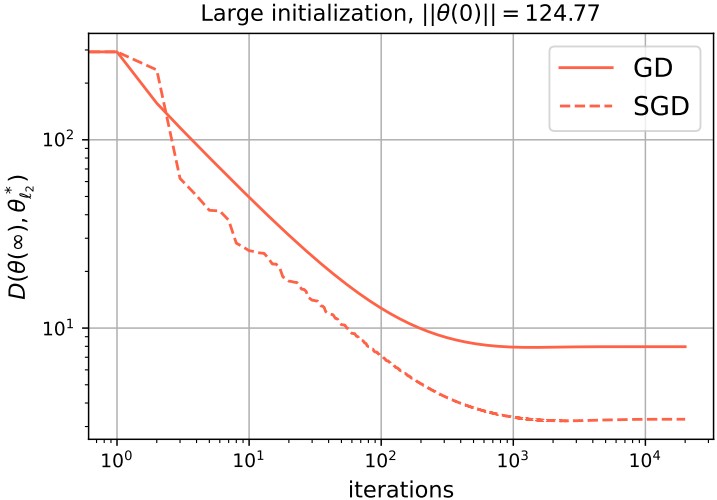

Figure 7: $D(\theta(t), \theta_{\ell_2}^*)$ along training for rank-1 linear networks when the initialization is extremely large. SGD solution is closer to $\theta_{\ell_2}^*$ when compared to the GD solution for rank-1 linear networks.

We now present the details of numerical experiments conducted in Section 5.

$\ell_2$ **minimum norm solution.** To get the $\ell_2$ minimum norm solution $\theta_{\ell_2}^*$, we train a single layer linear network with zero initialization using GD for 20000 iterations, since GD will return an $\ell_2$ minimum norm solution solution in this case according to Corollary 1.1 and [26].

**Rank-1 linear networks.** For the rank-1 linear network $f(x; u, v) = w_L^T W_{L-1} \cdots W_1 x$ where $W_k = u_k v_k^T \in \mathbb{R}^{d_k \times d_{k+1}}$, we let $L = 3$ and $\forall k \in \{1, \ldots, L\} : d_k = 100$. The learning rate is $10^{-3}$ and the batch size is 4 if we run SGD. We construct different rank-1 linear networks as follows: for a randomly sampled $\tilde{\theta} \in \mathbb{R}^{100}$, we let the initialization $\theta_i(0)$ of the $i$-th rank-1 networks have the same direction as $\tilde{\theta}$ but with different scales. We then train each rank-1 linear network with GD and SGD, respectively, for 20000 iterations. In particular:

1. Fig. 5(a) presents the results of the distances between $\theta_{\ell_2}^*$ and GD and SGD solutions, respectively, of each trained rank-1 networks with different initialization scales.

2. In Fig. 5(b), we measure the distances between $\theta^*$ and GD and SGD solutions, respectively, for each trained rank-1 networks with different initialization scales.

3. Fig. 6(a) plots the distances between $\theta^*_{\ell_2}$ and the model parameters $\theta$ along training when the initialization scales are different for both GD and SGD. The numbers in the bracket denote $\|\theta(0)\|$.

4. Fig. 7 is about the distances between $\theta^*_{\ell_2}$ and the model parameters $\theta$ along training for both GD and SGD when $\|\theta(0)\|$ is extremely large.

**Standard linear networks.** For the standard linear network $f(x;W) = w_L^T W_{L-1} \cdots W_1 x$ where $W_k \in \mathbb{R}^{d_k \times d_{k+1}}$, we let $L = 4$ and $\forall k \in \{1,2,3\} : d_k = 100$. The learning rate is $10^{-3}$ and the batch size is 4 if we run SGD. Other settings are similar to that of rank-1 linear networks.

1. In Fig. 5(c), we plot the distances between $\theta^*_{\ell_2}$ and GD and SGD solutions, respectively. Similar to the case of rank-1 linear networks, for all initialization scales, $D(\theta(\infty), \theta^*_{\ell_2})$ is smaller if the network is trained with SGD when compared to GD.

2. Fig. 5(d) presents the results of the distances between $\theta^*$ and GD and SGD solutions.

3. Fig. 6(b) plots the distances between $\theta^*_{\ell_2}$ and the model parameters $\theta$ along training when the initialization scales $\|\theta(0)\|$ are different for both GD and SGD. The numbers in the bracket denote $\|\theta(0)\|$.

**Non-linear networks.** For the non-linear network $f(x;W) = w_L^T \sigma(W_{L-1} \cdots \sigma(W_1 x))$ where $W_k \in \mathbb{R}^{d_k \times d_{k+1}}$, we let $L = 4$ and $\forall k \in \{1,2,3\} : d_k = 100$. The learning rate is $10^{-3}$ and the batch size is 4 if we run SGD. We use the ReLU activation $\sigma(x) = \mathrm{ReLU}(x)$. We use the same dataset as in the experiments of rank-1 linear networks. Since the non-linear networks do not have the overall parameterization of $\theta$ as in the linear networks case, to measure the initialization scale, we first straight all weight matrices to vectors and stack them to get a single vector, then we calculate the $\ell_2$ norm of this vector as the scale of the initialization of a non-linear network, i.e., we use $\sqrt{(\sum_k \|W_k(0)\|_F^2)}$ as the initialization scale where $\| \cdot \|_F$ is the Frobenius norm. Due to the same reason, we can not measure quantities such as $D(\theta, \theta^*_{\ell_2})$, therefore, we report the test error of the model in a newly sampled test set instead. Fig. 6(c) plots the test error of the model along training when the initialization scales are different for both GD and SGD. The numbers in the bracket denote initialization scales.

### A.2 Additional Experiments of Comparison with Diagonal Nets

Results in Section 3.2 indicate that diagonal linear networks exhibit different implicit bias in comparison with rank-1 and standard linear networks, e.g., both rank-1 and standard linear networks prefer $\ell_2$ minimum norm solution for GD when the initialization is nearly-zero while, on the contrary, diagonal linear networks prefer such solution when the initialization is sufficiently large. In this section, we empirically compare the implicit bias for rank-1 linear networks and diagonal linear networks to show this phenomenon.

In particular, we use the same settings as in the experiments for rank-1 linear networks in A.1 while only change the model to diagonal linear networks. As in previous works [3, 22], the re-parameterization of diagonal linear network is

$$\theta = w_+ \odot w_+ - w_- \odot w_-,$$

where $w_+ \in \mathbb{R}^{100}, w_- \in \mathbb{R}^{100}$ and $\odot$ is the elementwise product. Let $\mathbf{e} = (1, \cdots, 1)^T \in \mathbb{R}^{100}$, we set the initialization as

$$w_+(0) = C\mathbf{e}, \quad w_-(0) = C\mathbf{e},$$

where $C$ is a positive constant measuring the initialization scale. For each diagonal linear network with different $C$, we run GD for 20000 iterations and calculate $D(\theta(\infty), \theta^*_{\ell_2})$ and $D(\theta(\infty), \theta^*)$. The results are plotted in Fig. 8, where, for convenience of comparing the implicit bias of GD for rank-1 linear networks with that of diagonal linear networks, we also plot the results of rank-1 linear networks of Fig. 5(a) and Fig. 5(b) in Fig. 8(a) and Fig. 8(b), respectively.

As shown in Fig. 8(a), as the initialization scale ($\|\theta(0)\|$ for rank-1 linear networks, $C$ for diagonal linear networks) increases, $D(\theta(\infty), \theta^*_{\ell_2})$ decreases for rank-1 linear linear networks trained with

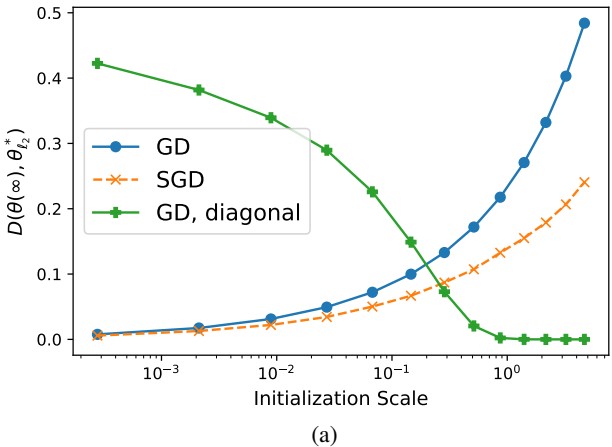

(a)

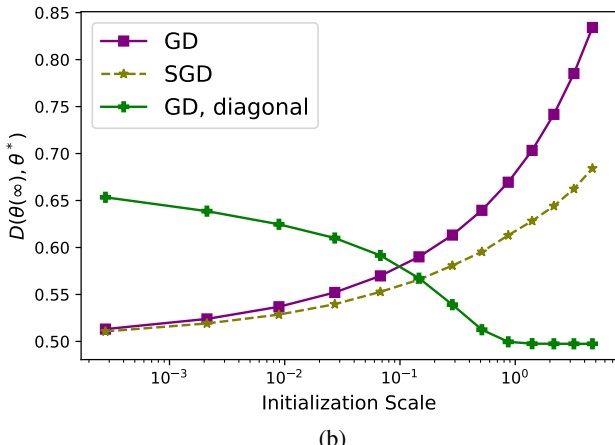

(b)

Figure 8: For different initialization scale ($\|\theta(0)\|$ for rank-1 linear networks and $C$ for diagonal linear networks): **(a)** $D(\theta(\infty), \theta^*_{\ell_2})$ for rank-1 linear nets and diagonal linear networks (the green solid line). **(b)** $D(\theta(\infty), \theta^*)$ for rank-1 linear nets and diagonal linear nets (the green solid line).

both GD and SGD, while it increases for diagonal linear networks trained with GD. This indicates the drastic difference between the implicit bias of GD exhibited by diagonal linear networks and rank-1 linear networks (also standard linear networks).

### A.3 Additional Experiments for the "Alleviating" Effect in Theorem 3

Recall the form of $V^{\mathcal{S}}(\theta, t)$ in Theorem 3

$$V^{\mathcal{S}}(\theta, t) = \frac{1}{\Omega_L} \|\theta\|^{\Omega_L} + \frac{\theta^T \theta(0)}{\|\theta(0)\|^{\lambda_L}} - \frac{2\lambda_L \eta \theta^T}{nb} \int_0^t \frac{\mathcal{L}(\theta(s)) \operatorname{tr}\left(P_\perp(\theta(s)) X^T X\right)}{\|\theta(s)\|^{2-\lambda_L}} \theta(s) ds,$$

we let

$$p_\theta(t) = \frac{\theta^T(t)\theta(0)}{\|\theta(0)\|^{\lambda_L}}, \tag{13}$$

$$q_\theta(t) = \frac{2\lambda_L \eta \theta^T(t)}{nb} \int_0^t \frac{\mathcal{L}(\theta(s)) \operatorname{tr}\left(P_\perp(\theta(s)) X^T X\right)}{\|\theta(s)\|^{2-\lambda_L}} \theta(s) ds. \tag{14}$$

To quantitatively measure the "alleviating" effect of the SGD sampling noise, we train 3 rank-1 linear networks with different initialization scales using SGD. The batch size is 4 and the learning rate

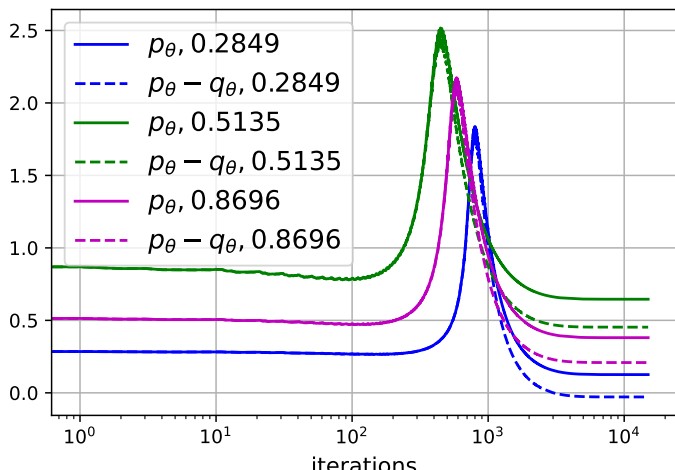

Figure 9: SGD sampling noise alleviates the dependence on the initialization. Numbers after the comma denote the initialization scales. The solid lines are for $p_\theta$ (Eq. (13)) and the dotted lines are for $p_\theta - q_\theta$ (Eq. (14)).

is $5 \times 10^{-5}$. For the rank-1 linear network $f(x; u, v) = w_L^T W_{L-1} \cdots W_1 x$ where $W_k = u_k v_k^T \in \mathbb{R}^{d_k \times d_{k+1}}$, we let $L = 3$ and $\forall k \in \{1, \ldots, L\} : d_k = 100$. We calculate both $p_\theta(t)$ and $q_\theta(t)$ along training, where $q_\theta(t)$ measures the alleviating effect of the SGD sampling noise and their difference $p_\theta(\infty) - q_\theta(\infty)$ is the alleviated initialization dependence of the SGD solution compared to GD solution.

As shown in Fig. 9, the effect coming from the SGD sampling noise, $q_\theta$, is equivalent to make the dependence of $V^S$ on the initialization closer to 0 (after about 1000 iterations, every dotted line is closer to the x-axis compared to the corresponding solid line with the same color), thus it controls the dependence of the SGD solution on the initialization. This phenomenon further verifies our claims.

### A.3.1 Training loss for Fig. 4(b)

Fig. 4(b) indicates that the final direction of the integral term in Eq. (12) highly depends on the initialization $\theta(0)$ since the loss decays along training. To further support this argument, here we present the training loss when we perform the experiments of Fig. 4(b) in Fig. 10. It can be seen that, for a random initialization, the loss, the magnitude of the speed of $\alpha$, has a high value at the start of the training, and decays very quickly, which explains why the direction of $\theta(0)$ is crucial to that of $\alpha(\infty)$.

### A.4 Additional Experiments for Biased Initialization

In this section, we provide additional experiments to show that our conclusion still holds when removing the balanced initialization condition (Definition 1).

To make the initialization unbalanced, we add a small perturbation to the balanced initialization. Specifically, we define

$$\Delta = \frac{1}{2L - 1} \sum_{k=1}^{L-1} \frac{|\|v_{k+1}\|^2 - \|u_k\|^2|}{\|u_k\|^2}$$

as the scale of the perturbation to the balanced initialization (larger $\Delta$ implies that the initialization is more unbalanced). All the other experiment details are kept unchanged as in Section 5. As shown in Fig. 11 and Fig. 12, we still observe similar phenomenons as in the case of the balanced initialization, e.g., SGD solutions are closer to the $\ell_2$-norm minimization solution compared to GD, when a small perturbation is added to the balanced initialization. Thus the implicit bias is not unique to the balanced initialization. In particular:

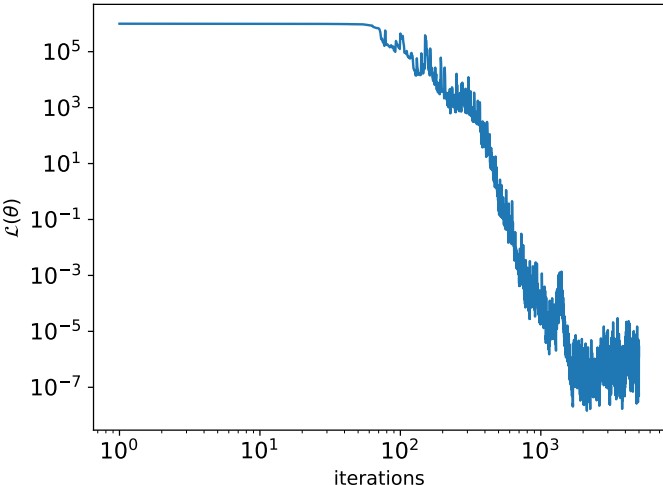

Figure 10: The empirical loss $\mathcal{L}(\theta)$ along training for Fig. 4(b).

- We report $D(\theta(\infty), \theta_{\ell_2}^*)$ for both GD and SGD for different levels of perturbation $\Delta$ (denoted in the title of each figure) in Fig. 11. In the last figure, we fix the initialization scale and report $D((\theta(\infty), \theta_{\ell_2}^*)$ of both GD and SGD for different $\Delta$. It can be seen that, without the balanced initialization, GD and SGD still prefer $\ell_2$-norm minimization solution $\theta_{\ell_2}^*$ for small initialization, while the SGD solution is closer to $\theta_{\ell_2}^*$ due to its initialization reduction effect.

- We report $D(\theta(t), \theta_{\ell_2}^*)$ during optimization for both GD and SGD for different levels of perturbation $\Delta$ and the same scale of initialization ($\|\theta(0)\| = 0.8696$) in Fig. 12, which further clearly reveals that there are still similar phenomenons when $\Delta \neq 0$ as in the case when the initialization is balanced.

## B   More Discussions

Our work proposes the rank-1 linear network which is a plausible proxy of standard linear networks with some neurons fully connected with neurons in its last and next layers. By showing that the proposed rank-1 linear networks are standard linear networks with special initialization, our conclusions may be generalized to standard linear networks. In comparison, the diagonal linear network, a special kind of linear networks that receives a lot of attention recently, does not have fully connected neurons. Furthermore, we find that the implicit bias of both GD and SGD for diagonal linear networks are not consistent with ours. The diagonal linear networks also exhibit drastically different implicit bias of GD when compared to standard linear networks, while the conclusions for rank-1 linear networks are consistent with those of standard linear networks. We also reveal the key role of the over-parameterization in characterizing the implicit bias of SGD, namely that it will only be different with that of GD for over-parameterization model.

The inconsistency between the implicit bias of GD and SGD for diagonal linear networks and rank-1 linear networks leads us to suggest intriguing questions for future work such as *what about other architectures* and *is there any unified analytical approach for studying implicit bias of GD and SGD for different architectures?* And it is interesting to reveal whether the "alleviating" initialization effect of the SGD sampling noise is general accross different architectures.

We precisely characterize the implicit bias of both GD and SGD for rank-1 linear networks, where the dependence on the initialization and depth is explicit and clear. In this sense, we take a step forward in the direction of characterizing the implicit bias of optimization algorithms.

Finally, our analysis characterizes the implicit bias of SGD through analyzing the overall parametrization $\theta$. This is different with another line of recent work [4, 10, 19, 27, 9] which focused on the

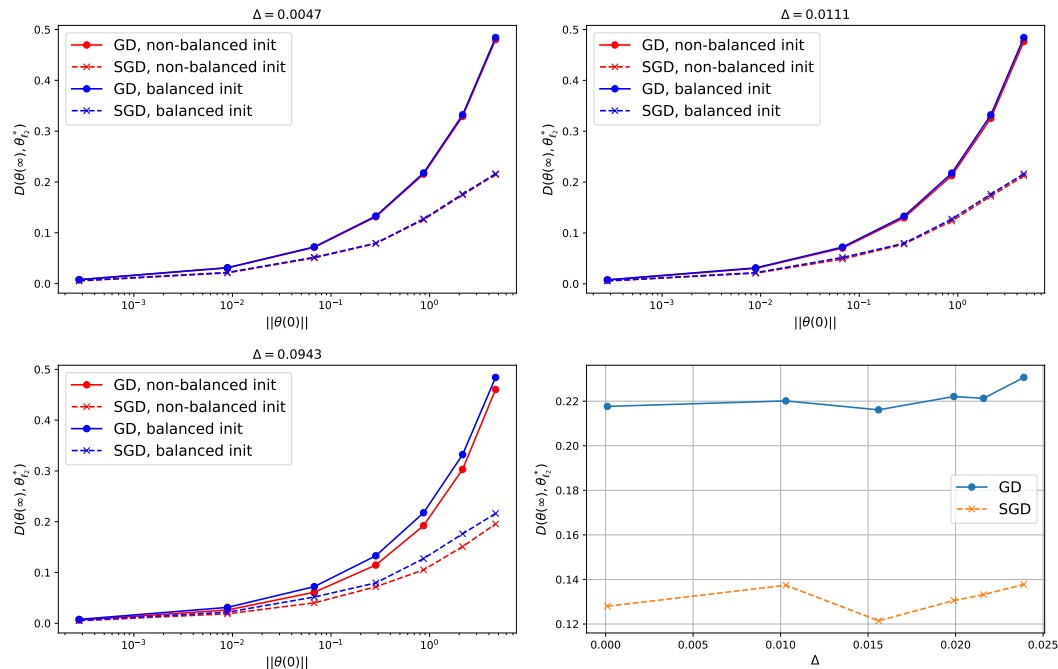

Figure 11: $D(\theta(\infty), \theta_{\ell_2}^*)$ for different $\|\theta(0)\|$ when the initialization is unbalanced ($\Delta \neq 0$, larger $\Delta$ means the initialization is more unbalanced). We use solid lines for the results of GD and dashed lines for SGD. For results under the balanced initialization, we use blue lines; for the results when a small perturbation is added to the balanced initialization, i.e., $\Delta \neq 0$, we use red lines.

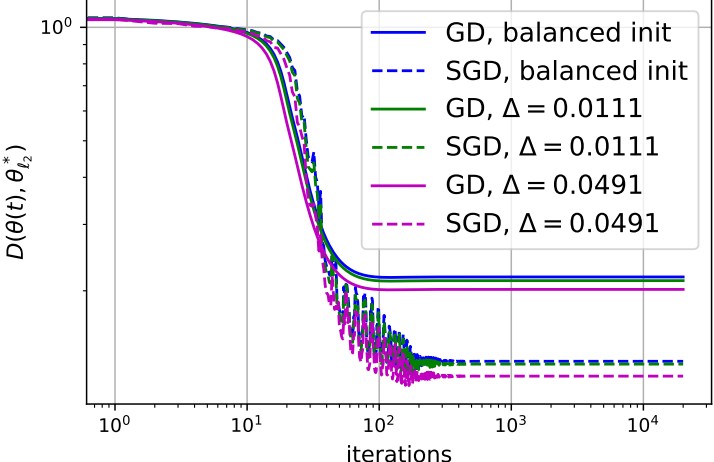

Figure 12: $D(\theta(t), \theta_{\ell_2}^*)$ along training for rank-1 linear networks when the initialization is unbalanced ($\Delta \neq 0$).

flatness of the loss landscape by directly analyzing the independent model parameters, i.e., $u$ and $v$ for the rank-1 linear networks, which is also crucial to fully understand the learning dynamics.

It is worth to mention that removing the balanced initialization (Definition 1) is also a promising direction. As verified by the numerical experiments in Appendix A.4, similar phenomena exist for unbalanced initialization. From the theoretical aspect, balanced initialization enables us to derive the exact dynamics of the overall parameter $\theta$, which is necessary to precisely characterize the implicit bias of GD/SGD. And it is difficult to discuss arbitrary initialization without the balanced initialization assumption. The effect of removing the balanced initialization is that the induced mirror flow potential should be composed of two parts: the original potential presented in Section 3 and a perturbation due to the imbalance of the initialization to it. This implies that the $\ell_2$-norm solution is still returned for small initialization. On the other hand, the case for SGD is much more complicated: the Brownian motion term of the corresponding SDE will also be affected by the imbalance of the initialization, which in turn induces a much more complex time varying mirror flow potential. We believe that the exact theoretical characterization of the implicit bias of SGD without the balanced initialization is a valuable future direction.

There are also some limitations in the current work. For example, although the conclusions of numerical experiments conducted on non-linear networks resemble that of rank-1 and standard linear networks, we can not directly generalize the current theoretical analysis to non-linear neural networks, which normally do not have overall parametrization vectors as $\theta$ that is necessary for our analysis. Moreover, the exact characterization of the stochastic integral is also absent in the current work, while we expect that the integral term $\mathcal{L}(\theta)\text{tr}\left(P_{\perp}(\theta)X^T X\right)\frac{\theta}{\|\theta\|}$ in Eq. (12) might have close relation with the property of the training dynamics.

## C    Proofs for Section 3

In this section, we present the technical details of Section 3. In particular, Section C.1 discusses the balanced initialization, Section C.2 proves Theorem 1 for rank-1 linear networks and Section C.3 proves Theorem 2 for standard linear networks.

For a rank-1 linear network $f(x; u, v) = w_L^T W_{L-1} \cdots W_1 x$ where $W_k = u_k v_k^T$ for any $k \in \{1, \ldots, L-1\}$, recalling the definition Eq. (3) and that the network $f(x; u, v)$ can be written as $f(x; u, v) = \rho_k v_{k+1}^T u_k \rho_{-k} v_1^T x$. For convenience, we let $\xi = w_L^T W_{L-1} \cdots W_2 u_1$.

### C.1    Balanced initialization

For a rank-1 linear network Eq. (2), dynamics of gradient flow is given by

$$\frac{du_k}{dt} = -\frac{2}{n}\rho_k \rho_{-k} v_1^T X^T r v_{k+1}, \tag{15}$$

$$\frac{dv_{k+1}}{dt} = -\frac{2}{n}\rho_k \rho_{-k} v_1^T X^T r u_k. \tag{16}$$

Based on this set of dynamics, we first discuss he following useful lemma that characterizes the dynamics of norms of model parameters:

**Lemma 1.** *For $f(x; u, v)$ trained with gradient flow, we have*

$$\forall k \in \{1, \ldots, L-1\} : \frac{d\|u_k\|^2}{dt} = \frac{d\|v_k\|^2}{dt} = \frac{d\|v_{k+1}\|^2}{dt}, \tag{17}$$

*i.e., layer norms grow at the same rate. Furthermore, if $\forall k \in \{1, \ldots, L-1\} : \|u_k(0)\| = \|v_{k+1}(0)\| = \|v_k(0)\|$, we have*

$$\forall k \in \{2, \ldots L-1\} : \frac{d\langle v_{k+1}, u_k\rangle^2}{dt} = \frac{d\langle v_k, u_{k-1}\rangle^2}{dt}. \tag{18}$$

*Proof.* Using Eq.(16), we have

$$\frac{1}{2}\frac{d\|u_k\|^2}{dt} = \left(\frac{du_k}{dt}\right)^T u_k = -\frac{2}{n}\rho_k\rho_{-k}v_1^T X^T r v_{k+1}^T u_k = -\frac{2}{n}\xi v_1^T X^T r, \tag{19}$$

$$\frac{1}{2}\frac{d\|v_{k+1}\|^2}{dt} = \left(\frac{du_k}{dt}\right)^T u_k = -\frac{2}{n}\rho_k\rho_{-k}v_1^T X^T r u_k^T v_{k+1} = -\frac{2}{n}\xi v_1^T X^T r. \tag{20}$$

Therefore, both $\frac{d\|u_k\|^2}{dt}$ and $\frac{d\|v_{k+1}\|^2}{dt}$ do not depend on $k$ and are same then Eq. (17) follows.

We now discuss Eq. (18). Since we assume $\forall k \in \{1, \ldots, L-1\} : \|u_k(0)\| = \|v_{k+1}(0)\| = \|v_k(0)\|$ and Eq. (17) implies that for any $t > 0$:

$$\|u_k(t)\|^2 - \|u_k(0)\|^2 = \|v_{k+1}(t)\|^2 - \|v_{k+1}(0)\|^2 = \|v_k(t)\|^2 - \|v_k(0)\|^2, \tag{21}$$

we have

$$\|u_k(t)\|^2 = \|v_{k+1}(t)\|^2 = \|v_k(t)\|^2 = \|u_1(t)\|^2 \tag{22}$$

To show Eq. (18), we note that

$$\begin{aligned}
\frac{d\langle v_{k+1}, u_k\rangle}{dt} &= \left(\frac{dv_{k+1}}{dt}\right)^T u_k + v_{k+1}^T \frac{du_k}{dt} \\
&= -\frac{2}{n}\rho_k\rho_{-k}v_1^T X^T r(\|u_k\|^2 + \|v_{k+1}\|^2) \\
&= -\frac{2}{n}\xi v_1^T X^T r\frac{\|u_k\|^2 + \|v_{k+1}\|^2}{\langle v_{k+1}, u_k\rangle} = -\frac{4}{n}\xi v_1^T X^T r\frac{\|u_1\|^2}{\langle v_{k+1}, u_k\rangle},
\end{aligned} \tag{23}$$

where we use Eq.(16) in the second equality and the third equality is because $\xi = \rho_k\rho_{-k}\langle v_{k+1}, u_k\rangle$. As a result, the above equation implies that

$$\frac{1}{2}\frac{d(\langle v_{k+1}, u_k\rangle)^2}{dt} = -\frac{4}{n}\xi v_1^T X^T r\|u_1\|^2, \tag{24}$$

which does not depend on $k$, and Eq. (18) follows. $\square$

To simplify the analysis, in Theorem 1, we have required the balanced initialization across layers (Definition 1). Recall that the balanced initialization is defined as

**Definition 1** (Balanced initialization for rank-1 linear networks). *Given an L-layer rank-1 linear network Eq. (2), for any $k \in \{1, \ldots, L-1\}$, the balanced initialization means that*

$$\frac{\langle v_{k+1}(0), u_k(0)\rangle^2}{\|v_{k+1}(0)\|^2\|u_k(0)\|^2} = 1, \tag{25}$$

$$\|v_{k+1}(0)\| = \|u_k(0)\| = \|v_1(0)\|. \tag{26}$$

Eq. (25) states that $u_k$ of the $k$-th layer is aligned with $v_{k+1}$ of the $(k+1)$-th layer in direction while Eq. (26) means they have the same magnitudes as $v_1(0)$. The balanced initialization has been suggested by several previous works [3, 2, 6, 28] for standard linear networks defined as follows.

**Definition 2** (Balanced initialization for standard linear networks)*. Given an L-layer standard linear network $f(x; W) = w_L^T W_{L-1} \cdots W_1 x$, for any $k \in \{1, \ldots, L-1\}$, the balanced initialization means that*

$$W_{k+1}^T(0)W_{k+1}(0) = W_k(0)W_k^T(0)$$

*for any $k \in \{1, \ldots, L\}$.*

This directly means that $W_{k+1}(0)$ and $W_k(0)$ share same singular values and $W_{k+1}$'s right singular vector aligns with the left singular vector of $W_k(0)$. In our case, such reasoning gives us

$$\frac{v_{k+1}(0)}{\|v_{k+1}(0)\|} = \frac{u_k(0)}{\|u_k(0)\|} \tag{27}$$

$$\|v_{k+1}(0)\|\|u_{k+1}(0)\| = \|v_k(0)\|\|u_k(0)\|, \tag{28}$$

where Eq. (27) is similar to Eq. (25), which shows that $u_k(0)$ aligns with $v_{k+1}(0)$, and we adapt the condition (28) to Eq. (26) for rank-1 linear net since $v_k$ and $u_k$ are the independent model parameters.

A nice property of GD is that the balanced property across layers will be maintained during training [6, 11, 2], i.e., $W_{k+1}^T(t)W_{k+1}(t) = W_k(t)W_k^T(t)$ for $t > 0$ and $k \in \{1, \dots, L\}$, which can be showed by taking derivative with respect to time on $W_{k+1}^T(t)W_{k+1}(t)$ and $W_k(t)W_k^T(t)$. For the rank-1 linear network case, according to Lemma 1 and Eq. (25), $\langle v_{k+1}(t), u_k(t) \rangle$ are the same for all $k \in \{1, \dots, L-1\}$, thus GD also maintains the balanced property for rank-1 deep linear networks. Although the balanced initialization conditions are slightly strict, it can be approximately accurate if the initialization scale is not large, which is rather common in practice. Under such initialization conditions, we are able to precisely characterize the implicit bias of GD and focus more on the effects coming from the overall initialization of model parameters, rather than the difference between layers.

## C.2 Proof of Theorem 1

In this section, we prove Theorem 1. Basically the idea is to show the existence of a potential function $V(\theta)$ such that the model parameter $\theta$ follows a mirror descent with respect to $V(\theta)$:

$$\theta(\infty) = \arg\min_\theta V(\theta) \quad s.t. \ X\theta = y. \tag{29}$$

This method is called infinitesimal mirror descent (IMD) approach and can be found in, e.g., [3]. Note that, to apply this method, the dynamics of $\theta$ should satisfy certain condition, which might be strict. For example, given the linear model $f(x; \theta) = \theta^T x$ for $\theta \in \mathbb{R}^d$, both parameterization of $\theta$ with standard linear networks, i.e. $\theta^T = w_L^T W_{L-1} \cdots W_1$, and a much simpler one $\theta = cv$ for $c \in \mathbb{R}$ and $v \in \mathbb{R}^d$ do not satisfy the condition of applying the IMD approach, while the rank-1 linear networks satisfy the condition, which implies that the parameterization of rank-1 linear networks is different with a scalar times a vector. Furthermore, the above example also confirms our motivation of studying rank-1 linear networks as a proxy of standard linear networks, especially considering that the implicit bias of SGD for rank-1 linear networks is more amenable.

We first present a useful Lemma in [20]:

**Lemma 2.** *If $H$ has rank 1 and $G$ is invertible, then*

$$(G + H)^{-1} = G^{-1} - \frac{1}{1+g} G^{-1} H G^{-1}. \tag{30}$$

*where $g = \mathrm{tr}\left(HG^{-1}\right)$.*

We now prove Theorem 1.

*Proof.* Recall that $r \in \mathbb{R}^n$ with $r_i = (f_i - y_i)$, we let

$$\Phi = \sum_{k=1}^{L-1} \phi_k, \tag{31}$$

$$\phi_k = \rho_k^2 \rho_{-k}^2 (\|u_k\|^2 + \|v_{k+1}\|^2). \tag{32}$$

The key step is to derive the dynamics of the overall parameter $\theta$ which can be done by noting that

$$\frac{d\theta}{dt} = v_1 \frac{d\xi}{dt} + \xi \frac{dv_1}{dt},$$

where, according to Lemma 1,

$$\frac{d\xi}{dt} = \sum_{k=1}^{L-1} \rho_k \rho_{-k} \frac{d \langle v_{k+1}, u_k \rangle}{dt}$$

$$= -\frac{2}{n} r^T X v_1 \sum_{k=1}^{L-1} \rho_k^2 \rho_{-k}^2 (\|u_k\|^2 + \|v_{k+1}\|^2)$$

$$= -\frac{2}{n} \Phi v_1^T X^T r \tag{33}$$

$$\frac{dv_1}{dt} = -\frac{2}{n} \xi X^T r \tag{34}$$

$$\implies \frac{d\theta}{dt} = -\frac{2}{n} \left( \xi^2 I + \Phi v_1 v_1^T \right) X^T r. \tag{35}$$

Eq. (35) can be rewritten as

$$\left(\xi^2 I + \Phi v_1 v_1^T\right)^{-1} \frac{d\theta}{dt} = -\frac{2}{n} X^T r. \tag{36}$$

According to Lemma 2, note that

$$\mathrm{tr}\left(\Phi \frac{v_1 v_1^T}{\xi^2}\right) = \frac{\Phi \|v_1\|^2}{\xi^2},$$

the inverse appeared in Eq. (36) is

$$\left(\xi^2 I + \Phi v_1 v_1^T\right)^{-1} = \frac{1}{\xi^2} I - \frac{\xi^{-2} v_1 v_1^T \Phi \xi^{-2}}{1 + \frac{\Phi \|v_1\|^2}{\xi^2}} = \frac{1}{\xi^2} I - \frac{\theta \theta^T}{\frac{\xi^6}{\Phi} + \xi^2 \|\theta\|^2}. \tag{37}$$

It is now left for us to express $\xi^2$ and $\Phi$ in terms of $\theta$. In the following, we assume the balanced initialization in Theorem 1 and apply Lemma 1.

1. $\xi^2$. This can be done by noting that $\|\theta\|^2 = \xi^2 \|v_1\|^2$, where $\xi^2$ is given by

$$\xi^2 = \prod_{k=1}^{L-1} \langle v_{k+1}, u_k \rangle^2. \tag{38}$$

   Note that $\langle v_{k+1}, u_k \rangle^2$ grow at the same rate for different $k$ according to Lemma 1 and $\langle v_{k+1}, u_k \rangle^2$ are the same at initialization for different $k$ due to our assumption, we have $\langle v_{k+1}, u_k \rangle^2 = \langle v_2, u_1 \rangle^2$ and $\xi^2 = \langle v_2, u_1 \rangle^{2(L-1)}$. Note that

$$\frac{1}{2} \frac{d \langle v_2, u_1 \rangle^2}{dt} = -\frac{4}{n} \xi v_1^T X^T r \|u_1\|^2 = -\frac{4}{n} \xi v_1^T X^T r \|v_1\|^2 = \frac{1}{2} \frac{d \|v_1\|^4}{dt}, \tag{39}$$

   we have $\langle v_2, u_1 \rangle^2 - \langle v_2(0), u_1(0) \rangle^2 = \|v_1\|^4 - \|v_1(0)\|^4$. Since we have $\langle v_2, u_1 \rangle^2 = \|u_1\|^4 = \|v_1\|^4$ at initialization according to our assumption, $\xi^2$ can be finally written as

$$\xi^2 = \|v_1\|^{4(L-1)}. \tag{40}$$

   As a result,

$$\|\theta\| = \|v_1\|^{2L-1} \implies \|v_1\| = \|\theta\|^{\frac{1}{2L-1}}, \quad \xi^2 = \|\theta\|^{\frac{4(L-1)}{2L-1}}. \tag{41}$$

2. $\xi^6/\Phi$. By taking some simple algebra, we have

$$\Phi = \frac{2(L-1)\xi^2}{\|v_1\|^2} \implies \frac{\xi^6}{\Phi} = \frac{\xi^4 \|v_1\|^2}{2(L-1)}. \tag{42}$$

Now Eq. (36) becomes

$$\|\theta\|^{-\frac{2(L-1)}{2L-1}} \left(I - \frac{\theta \theta^T}{\frac{\|\theta\|^2}{2(L-1)} + \|\theta\|^2}\right) \frac{d\theta}{dt} = -\frac{2|\xi|}{n} X^T r. \tag{43}$$

These conditions are now sufficient for us to find the form of the potential $V(\theta)$. Suppose that $V(\theta)$ can be written as

$$V(\theta) = \hat{V}(\|\theta\|) + h^T \theta \tag{44}$$

for some vector $h$ and satisfies the following relation:

$$\nabla_\theta^2 V(\theta) = \nabla_\theta^2 \hat{V}(\theta)$$

$$= \|\theta\|^{-\frac{2(L-1)}{2L-1}} \left(I - \frac{1}{1 + \frac{1}{2(L-1)}} \frac{\theta \theta^T}{\|\theta\|^2}\right), \tag{45}$$

then Eq. (43) gives us

$$\frac{d}{dt}\left(\nabla_\theta V(\theta)\right) = -\frac{2}{n} X^T r \tag{46}$$

and the integration relation

$$\nabla_\theta V(\theta) - \nabla_\theta V(\theta)|_{\theta=\theta(0)} = \sum_{i=1}^n x_i \int \tilde{r}_i(\tau) d\tau \tag{47}$$

where we let $\tilde{r} = -2|\xi|r/n$. Requiring $\nabla_\theta V(\theta)|_{\theta=\theta(0)} = 0$ and denoting $\lambda_i = \int_0^\infty \tilde{r}_i(\tau) d\tau$ gives us the condition at $t = \infty$:

$$\nabla_\theta V(\theta)|_{\theta=\theta(\infty)} = \sum_{i=1}^n x_i \lambda_i. \tag{48}$$

Eq. (48) coincides with the KKT stationary condition of the optimization problem (29). Therefore, we can prove the theorem by deriving the explicit form of $V(\theta)$.

**Solving $V(\theta)$.** According to Eq. (44), we can derive the following relation:

$$\partial_\theta V(\theta) = \hat{V}' \frac{\theta}{\|\theta\|} + h^T \tag{49}$$

$$\partial_\theta^2 V(\theta) = \frac{1}{\|\theta\|^2} \left[ \left( \hat{V}'' \frac{\theta\theta^T}{\|\theta\|} + \hat{V}'I \right) \|\theta\| - \hat{V}' \frac{\theta\theta^T}{\|\theta\|} \right]$$

$$= \frac{\hat{V}'}{\|\theta\|} \left[ I - \left( 1 - \|\theta\| \frac{\hat{V}''}{\hat{V}'} \right) \frac{\theta\theta^T}{\|\theta\|^2} \right]. \tag{50}$$

Comparing this with Eq. (45), we conclude that

$$1 - \|\theta\| \frac{\hat{V}''}{\hat{V}'} = \frac{1}{1 + \frac{1}{2(L-1)}} \implies \frac{\hat{V}''}{\hat{V}'} = \frac{1}{\|\theta\|} \left( 1 - \frac{1}{1 + \frac{1}{2(L-1)}} \right), \tag{51}$$

which, by noting that $\partial_{\|\theta\|} \ln \hat{V}'(\|\theta\|) = \frac{\hat{V}''(\|\theta\|)}{\hat{V}'(\|\theta\|)}$, can be solved as follows

$$\frac{\hat{V}''(\|\theta\|)}{\hat{V}'(\|\theta\|)} = \frac{1}{\|\theta\|} \frac{1}{2L-1}$$

$$\implies \ln \hat{V}'(\|\theta\|) = \frac{1}{2L-1} \ln \|\theta\|$$

$$\implies \hat{V}(\|\theta\|) = \frac{2L-1}{2L} \|\theta\|^{\frac{2L}{2L-1}}. \tag{52}$$

Furthermore, since

$$\frac{\hat{V}'}{\|\theta\|} = \|\theta\|^{-\frac{2(L-1)}{2L-1}},$$

Eq. (45) is automatically satisfied when $\hat{V}(\|\theta\|)$ has the form of Eq. (51). It is now left for us to get the form of $h$, which can be done by noting that

$$\partial_\theta V(\theta(0)) = 0$$

$$\implies \|\theta(0)\|^{\frac{1}{2L-1}} \frac{\theta(0)}{\|\theta(0)\|} + h = 0.$$

Thus the final form of $V(\theta)$ is

$$V(\theta) = \frac{1}{\Omega_L} \|\theta\|^{\Omega_L} - \theta^T \frac{\theta(0)}{\|\theta(0)\|^{\lambda_L}}. \tag{53}$$

This completes the proof. □

### C.2.1 Remove the assumption of convergence to the interpolation solution

The assumption that $X\theta(\infty) = y$ in Theorem 1 can be removed if the dimension of the span of $X^T$ is larger than the number of samples $n$, i.e., when $\dim\left(\text{span}(X^T)\right) \geq n$.[1] This can be proved as follows.

---

[1] Since $\max(\dim(\text{span}(X^T))) = n$, this condition is in fact $\dim(\text{span}(X^T)) = n$.

**Proposition 2.** *For the over-parameterized regression of rank-1 linear networks Eq. (2) and the dataset $\{(x_i, y_i)\}_{i=1}^n$ where $x_i \in \mathbb{R}^d$ and $n < d$, if*

$$dim\left(span(X^T)\right) \geq n,$$

*then the gradient flow solution $\theta(\infty)$ satisfies that*

$$X\theta(\infty) = y.$$

*Proof.* To prove this proposition, we study the dynamics of the loss function $\mathcal{L} = \frac{1}{n}\sum_i r_i^2$, where $r_i = \langle \theta, x_i \rangle - y_i$, that is given by

$$
\begin{aligned}
\frac{d\mathcal{L}}{dt} &= \frac{\partial \mathcal{L}}{\partial \theta}\frac{d\theta}{dt} \\
&= \frac{2}{n}r^T X \left[ -\frac{2\xi^2}{n}\left( I + 2(L-1)\frac{\theta\theta^T}{\|\theta\|^2} \right) X^T r \right] \\
&= -\frac{4\xi^2}{n^2}\left[ rX^T X r + 2(L-1)\frac{r^T X \theta\theta^T X^T r}{\|\theta\|^2} \right] \\
&= -\frac{4\xi^2}{n^2}\left[ \|X^T r\|^2 + 2(L-1)\frac{(\theta^T X^T r)^2}{\|\theta\|^2} \right] \\
&\leq 0,
\end{aligned}
$$

where we have the equality in the last line when $r = (0, \ldots, 0)^T \in \mathbb{R}^d$, i.e., $X\theta = y$, or $X^T r = 0$, which is not possible since we have assumed that $dim(span(X^T)) \geq n$. Therefore, $\mathcal{L}(\theta(t))$ keeps decreasing until $X\theta(t) = y$, i.e., until GD finds the interpolation solution. Noting that $\min_\theta \mathcal{L}(\theta) = 0$, we complete the proof. $\qquad\square$

### C.3 Proof of Theorem 2

In this section we prove Theorem 2. The techniques are similar to those in C.2, and we still need a time wrapping technique introduced in [3] to derive the form of $V_{std}(\theta)$, since the condition for applying the IMD approach is violated in this case. Recall that, for the standard linear networks $f(x; W) = w_L^T W_{L-1} \cdots W_1 x = \theta^T x$ where $W_k \in \mathbb{R}^{d_k \times d_{k+1}}$, our purpose is to find a potential function $V_{std}(\theta)$ such that the gradient flow solution $\theta(\infty)$ satisfies that

$$\theta(\infty) = \arg\min_\theta V_{std}(\theta), \quad s.t. X\theta = y. \tag{54}$$

Since we assume the balanced initialization (Definition 2), then according to [11, 6, 2], the norms of all layers grow at the same rate and are the same for any $t > 0$:

$$W_{k+1}^T(t)W_{k+1}(t) = W_k(t)W_k^T(t).$$

Furthermore, following the procedure of [2], we obtain that the dynamics of $\theta$ is

$$\frac{d\theta}{dt} = -\|\theta\|^{\frac{2(L-1)}{L}}\left[ I + (L-1)\frac{\theta\theta^T}{\|\theta\|^2} \right] X^T r. \tag{55}$$

We now present the proof.

*Proof.* Eq. (55) can be written as

$$\|\theta\|^{\frac{2(1-L)}{L}}\left[ I + (L-1)\frac{\theta\theta^T}{\|\theta\|^2} \right]^{-1}\frac{d\theta}{dt} = -X^T r,$$

where, according to Lemma 2, the inverse in above equation is given by

$$
\begin{aligned}
\left[ I + (L-1)\frac{\theta\theta^T}{\|\theta\|^2} \right]^{-1} &= I - \frac{1}{1 + \text{tr}\left( \frac{(L-1)\theta\theta^T}{\|\theta\|^2} \right)}(L-1)\frac{\theta\theta^T}{\|\theta\|^2} \\
&= I - \frac{L-1}{L}\frac{\theta\theta^T}{\|\theta\|^2}. \tag{56}
\end{aligned}
$$

As a result, we have

$$\|\theta\|^{\frac{2(1-L)}{L}} \left( I - \frac{L-1}{L} \frac{\theta\theta^T}{\|\theta\|^2} \right) \frac{d\theta}{dt} = -X^T r. \tag{57}$$

In this case, we still assume that $V_{std}(\theta)$ has the following form

$$V_{std}(\theta) = \hat{V}_{std}(\|\theta\|) + \beta^T \theta \tag{58}$$

for some constant vector $\beta \in \mathbb{R}^d$. Then following a similar procedure as in C.2, we have that

$$\partial_\theta V_{std}(\theta) = \hat{V}'_{std} \frac{\theta}{\|\theta\|} + h^T \tag{59}$$

$$\partial_\theta^2 V_{std}(\theta) = \frac{\hat{V}'_{std}}{\|\theta\|} \left[ I - \left( 1 - \|\theta\| \frac{\hat{V}''_{std}}{\hat{V}'_{std}} \right) \frac{\theta\theta^T}{\|\theta\|^2} \right]. \tag{60}$$

In the IMD approach, $V_{std}(\theta)$ should satisfy that

$$\frac{d}{dt}(\partial_\theta V_{std}(\theta)) = -\frac{2}{n} X^T r, \tag{61}$$

which requires that $\partial_\theta^2 V_{std}(\theta) = \|\theta\|^{\frac{2(L-1)}{L}} \left( I - \frac{L-1}{L} \frac{\theta\theta^T}{\|\theta\|^2} \right) \frac{d\theta}{dt}$. But this is not possible. Therefore, we multiply a time re-scale factor $g(\theta)$, as long as $g(\theta)$ is positive, to both sides of Eq. (57) and only require that $V_{std}(\theta)$ satisfies the above relation under the new time scale $\tau : \mathbb{R} \to \mathbb{R}$ such that $\tau' = g(\theta)$:

$$g(\theta)\|\theta\|^{\frac{2(1-L)}{L}} \left( I - \frac{L-1}{L} \frac{\theta\theta^T}{\|\theta\|^2} \right) \frac{d\theta}{dt} = -g(\theta)X^T r. \tag{62}$$

Then the limit point at $t = \infty$ in Eq. (61) is also visited at the point $\tau = \int_0^\infty g(\theta(s))ds$ in Eq. (62). We now solve the explicit form of $V_{std}(\theta)$ that satisfies Eq. (62). By comparing Eq. (60) and the right hand side of Eq. (57), we obtain that the following relation should be satisfied:

$$\frac{\hat{V}'_{std}}{\|\theta\|} \left[ I - \left( 1 - \|\theta\| \frac{\hat{V}''_{std}}{\hat{V}'_{std}} \right) \frac{\theta\theta^T}{\|\theta\|^2} \right] = g(\theta)\|\theta\|^{\frac{2(1-L)}{L}} \left( I - \frac{L-1}{L} \frac{\theta\theta^T}{\|\theta\|^2} \right). \tag{63}$$

This implies that we need:

- the terms in the bracket on both sides should match:

$$1 - \|\theta\| \frac{\hat{V}''_{std}}{\hat{V}'_{std}} = \frac{L-1}{L} \implies \frac{1}{xL} = \frac{\hat{V}''_{std}}{\hat{V}'_{std}}$$

$$\implies \ln \hat{V}'_{std} = \frac{1}{L} \ln \|\theta\| + C \implies \hat{V}_{std} = \frac{C'L}{L+1}\|\theta\|^{\frac{1}{L}+1} \tag{64}$$

  for some constant $C$ and $C'$, where we can simply choose $C' = 1$;

- the terms outside the bracket on both sides should also match:

$$\frac{\hat{V}'_{std}}{\|\theta\|} = g(\theta)\|\theta\|^{\frac{2(1-L)}{L}} \implies g(\theta) = \hat{V}'_{std}\|\theta\|^{\frac{2(L-1)}{L}-1}. \tag{65}$$

To obtain the form of the constant vector $\beta$, we note that $\partial_\theta V(\theta(0)) = 0$, which immediately gives us

$$\|\theta(0)\|^{\frac{1}{L}} \frac{\theta(0)}{\|\theta(0)\|} + \beta = 0 \implies \beta = -\theta(0)\|\theta(0)\|^{\frac{1}{L}-1}. \tag{66}$$

Combining all these terms, we have the final form of $V_{std}(\theta)$:

$$V_{std}(\theta) = \frac{L}{L+1}\|\theta\|^{\frac{1}{L}+1} - \theta(0)^T\theta\|\theta(0)\|^{\frac{1}{L}-1}. \tag{67}$$

As in C.2, denoting $\lambda_i = \int_0^\infty r_i(s)ds$ and noting that $\partial_\theta V_{std}(\theta(0)) = 0$ give us

$$\partial_\theta V_{std} = \sum_{i=1}^n x_i\lambda_i,$$

which is exactly the KKT stationary condition of the optimization problem (54). □

**Convergence to the interpolation solution.** Similar to C.2.1, we can also show the convergence to the interpolation solution when $\dim\left(\text{span}(X^T)\right) \geq n$ by deriving the dynamics of $\mathcal{L}$:

$$\frac{d\mathcal{L}}{dt} = \frac{\partial \mathcal{L}}{\partial \theta}\frac{d\theta}{dt}$$

$$= -\frac{2\|\theta\|^{\frac{2(1-L)}{L}}}{n} r^T X \left[I + (L-1)\frac{\theta\theta^T}{\|\theta\|^2}\right] X^T r$$

$$= -\frac{2\|\theta\|^{\frac{2(1-L)}{L}}}{n} \left[\|r^T X\|^2 + (L-1)\left(\theta^T X^T r\right)^2\right].$$

Since we assume that $\dim\left(\text{span}(X^T)\right) \geq n$, $d\mathcal{L}/dt < 0$ until $X\theta(t) = y$, i.e., $\mathcal{L}$ keeps decreasing until GD finds the interpolation solution.

## C.4 Proof of Proposition 1

In this section, we prove Proposition 1 by analyzing the gradient of model parameters. It is helpful to recall that for a matrix $A$, we use $A_{ij}$ to denote its $i$-th row $j$-th column element. For weight matrices, e.g., $W_k$, we use $W_{k;ij}$ to denote its $i$-th row $j$-th column element.

*Proof.* For a standard linear network that has the initialization of Proposition 1 and $L$ is an odd number, i.e., for an integer $p$

$$\forall k = 2p+1 \in \{1, \ldots, L\} : W_{k;ij}(0) = 0 \text{ if } i \neq c_k, \text{ where } c_k \in \{1, \ldots d_{k+1}\}$$
$$\forall k = 2p \in \{1, \ldots, L\} : W_{k;ij}(0) = 0 \text{ if } j \neq c_k,$$

note that the $L-1$-th layer satisfies that $W_{L-1;ij} = W_{L-1;ij}\delta_{jc_{L-1}}$ where $\delta_{jl} = 1$ if $j = l$ otherwise $\delta_{jl} = 0$, we can write the networks at $t = 0$ as

$$f(x; W) = \sum_i \sum_j w_{L;i} W_{L-1;ij}\delta_{jc_{L-1}} \left(W_{L-2}\cdots W_1 x\right)_j.$$

Then the gradient w.r.t $W_{L-1}$ at $t = 0$ is

$$\left(\nabla_{W_{L-1}}\mathcal{L}(\theta)\right)_{ij} = \frac{2}{n}\sum_{\mu=1}^n r_\mu w_{L;i}(W_{L-2}\cdots W_1 x)_j\delta_{jc_{L-1}} \tag{68}$$

$$\implies \left(\nabla_{W_{L-1}}\mathcal{L}(\theta)\right)_{ij} = 0 \text{ if } j \neq c_{L-1}. \tag{69}$$

This means that the parameter $W_{L-1;ij}$ will be updated only when $j = c_{L-1}$. As a result, the initialization shape for $W_{L-1}$ will be maintained, i.e., only the non-zero column of $W_{L-1}$ at $t = 0$ will be updated and all other elements of $W_{L-1}$ will be zero for any $t > 0$ since the corresponding gradients vanish. Similarly, for $W_{L-1}$, we can write the network at $t = 0$ as:

$$f(x; W) = \sum_j \sum_l (w_L^T W_{L-1})_j \delta_{jc_{L-1}} W_{L-2;jl}(W_{L-3}\cdots W_1 x)_l,$$

then

$$\left(\nabla_{W_{L-2}}\mathcal{L}(\theta)\right)_{ij} = 0 \text{ if } i \neq c_{L-1}$$

and the initialization shape for $W_{L-2}$ will also be maintained. Following a similar procedure, we conclude that all the initialization shapes will be maintained for any $t > 0$.

We now consider the diagonal initialization for weight matrices, namely that

$$W_{k;ij} = 0 \text{ if } i \neq j, \quad \forall k \in \{1, \ldots, L-1\}.$$

For any $k \in \{1, \ldots, L-1\}$, the gradient w.r.t $W_k$ at $t = 0$ is

$$\left(\nabla_{W_k}\mathcal{L}(\theta)\right)_{ij} = \frac{2}{n}\sum_{\mu=1}^n r_\mu(w_L^T\cdots W_{k+1})_i(W_{k-1}\cdots W_1 x)_j,$$

where, clearly, there is not any constraint on which elements of $(w_L^T\cdots W_{k+1})^T \in \mathbb{R}^d$ and $(W_{k-1}\cdots W_1 x) \in \mathbb{R}^d$ are zeros if we do not require that many diagonal elements of the initialization

matrices are zeros, which clearly violates the diagonal initialization requirements. Thus we can not conclude that

$$(\nabla_{W_k}\mathcal{L}(\theta))_{ij} = 0 \text{ if } i \neq j,$$

i.e., the off-diagonal elements of $W_k$ will also be updated to be non-zeros. Thus the diagonal initialization of weight matrices can not be maintained. The conclusions for other layers can be easily derived by following similar arguments. $\qquad\square$

# D   Proofs for Section 4

In this section, we present the technical details for Section 4. In particular, we introduce our modelling details of the SGD dynamics in D.1, derive the SDE of the model parameter $\theta$ in D.2, and, finally, prove Theorem 3 in D.3.

## D.1   SDE modelling details

Similar to the case of GD, to derive the implicit bias of SGD for rank-1 linear networks, we need to first derive the dynamics of the overall model parameter $\theta$. For this purpose, the structure of the noise of SGD is crucial. For convenience, we first discuss the basic SGD where only one data is randomly sampled at each step, while the generalization to batch-size SGD is straightforward.

**Structure of the noise.**   We present the details for $u_k$, and the case for $v_{k+1}$ is similar. Recall that the empirical loss is $\mathcal{L} = \sum_i \ell_i$ / n, $\eta$ is the learning rate and that the network can be written as $f(x; u, v) = \rho_k \rho_{-k} v_{k+1}^T u_k v_1^T x$, we start with the SGD update equation for $u_k$ where we let $j_t$ denote the index of the sampled data at the $t$-th step:

$$
\begin{aligned}
u_k(t+1) &= u_k(t) - \eta\frac{\partial\ell_{j_t}}{\partial u_k} = u_k(t) - \eta\frac{\partial\mathcal{L}}{\partial u_k} + \eta\left(\frac{\partial\mathcal{L}}{\partial u_k} - \frac{\partial\ell_{j_t}}{\partial u_k}\right) \\
&= u_k(t) - \frac{2\eta}{n}v_1^T X^T r\rho_k\rho_{-k}v_{k+1} \\
&\quad + 2\eta\left(\frac{1}{n}\sum_i r_i\rho_k\rho_{-k}v_1^T x_i v_{k+1} - r_{j_t}\rho_k\rho_{-k}v_1^T x_{j_t}v_{k+1}\right) \\
&= u_k(t) - \frac{2}{n}v_1^T X^T r\rho_k\rho_{-k}v_{k+1} + 2\eta\rho_k\rho_{-k}v_{k+1}v_1^T X^T Z_{j_t}
\end{aligned}
\tag{70}
$$

where we let $\overrightarrow{e_{j_t}}$ be the basis vector in $\mathbb{R}^n$ such that the $j_t$-th element is 1 while all other elements are 0 and

$$Z_{j_t} = \mathrm{E}_{j_t}[r_{j_t}\overrightarrow{e_{j_t}}] - r_{j_t}\overrightarrow{e_{j_t}}, \quad \mathrm{cov}[Z_{j_t}] \sim \frac{\mathcal{L}}{n}\mathbf{I}_n. \tag{71}$$

As a result of this, the noise of SGD for $u_k$ is now

$$\Sigma(u_k(t)) = \frac{4\eta^2\mathcal{L}}{n}(\rho_k\rho_{-k})^2 v_1^T X^T X v_1 v_{k+1}v_{k+1}^T. \tag{72}$$

**Continuous Modelling of SGD.**   The continuous modelling techniques for SGD have been widely applied in recent works [1, 9, 23, 22] to study the dynamics of SGD. In our setting, the continuous counterpart of SGD is established as follows. First, the discrete SGD updating equations Eq. (8) can be equivalently written as

$$
\begin{aligned}
u_k(t+1) &= u_k(t) - \eta\nabla_{u_k}\mathcal{L}(\theta) + \eta\left[\nabla_{u_k}\mathcal{L}(\theta) - \nabla_{u_k}\ell_{i(t)}(\theta)\right], \\
v_{k+1}(t+1) &= v_{k+1}(t) - \eta\nabla_{v_{k+1}}\mathcal{L}(\theta) + \eta\left[\nabla_{v_{k+1}}\mathcal{L}(\theta) - \nabla_{v_{k+1}}\ell_{i(t)}(\theta)\right]
\end{aligned}
$$

for $k \in \{1,\ldots,L-1\}$ where both $\nabla_{u_k}\mathcal{L}(\theta) - \nabla_{u_k}\ell_{i(t)}(\theta)$ and $\nabla_{v_{k+1}}\mathcal{L}(\theta) - \nabla_{v_{k+1}}\ell_{i(t)}(\theta)$ are zero-mean noises with covariance matrices in $\mathbb{R}^{d_k \times d_k}$

$$\Sigma(u_k) = \frac{4\mathcal{L}(\rho_k\rho_{-k})^2}{n}v_1^T X^T X v_1 v_{k+1}v_{k+1}^T,$$

$$\Sigma(v_{k+1}) = \frac{4\mathcal{L}(\rho_k\rho_{-k})^2}{n}v_1^T X^T X v_1 u_k u_k^T,$$

with $\rho_k$ and $\rho_{-k}$ defined in Eq. (3). Second, as we identify the noise covariance, letting $\eta \to 0$ [2], then we obtain the continuous counterpart of the discrete SGD that is a set of stochastic differential equations (SDE):

$$du_k = -\frac{2}{n}v_1^T X^T r \rho_k \rho_{-k} v_{k+1} dt + 2\sqrt{\frac{\eta\mathcal{L}}{n}}(\rho_k\rho_{-k})v_{k+1}v_1^T X^T d\mathcal{W}_t \tag{73}$$

$$dv_{k+1} = -\frac{2}{n}v_1^T X^T r \rho_k \rho_{-k} u_k dt + 2\sqrt{\frac{\eta\mathcal{L}}{n}}(\rho_k\rho_{-k})u_k v_1^T X^T d\mathcal{W}_t \tag{74}$$

where we let $r = (f(x_1;u,v) - y_1, \ldots, f(x_n;u,v) - y_n)^T \in \mathbb{R}^n$ be the residuals and $\mathcal{W}_t$ is a standard Brownian motion in $\mathbb{R}^n$. Similarly, recalling the definition of $\xi = w_L^T W_{L-1} \cdots W_2 u_1$, the SDE of $v_1$ is

$$dv_1 = -\frac{2}{n}\xi X^T r dt + 2\sqrt{\frac{\eta\mathcal{L}}{n}}\xi X^T d\mathcal{W}_t. \tag{75}$$

**Generalization to batch-SGD.** When $b$ (a positive constant) data points $\mathcal{B}_t$ are sampled in each iteration of SGD, i.e., batch-SGD, we can change the SGD update equation as follows (taking $u_k$ as an example)

$$u_k(t+1) = u_k(t) - \frac{\eta}{b}\sum_{j_t \in \mathcal{B}_t}\frac{\partial \ell_{j_t}}{\partial u_k}.$$

This only changes the noise $\Sigma(u_k(t))$ to a batch version

$$\Sigma_b(u_k(t)) = \frac{1}{b}\Sigma(u_k(t)),$$

which only affects the noise part of the SDE of $u_k$ and leads it to become a batch version SDE:

$$du_k = -\frac{2}{n}v_1^T X^T r \rho_k \rho_{-k} v_{k+1} dt + 2\sqrt{\frac{\eta\mathcal{L}}{nb}}(\rho_k\rho_{-k})v_{k+1}v_1^T X^T d\mathcal{W}_t.$$

This is equivalent to re-scale the learning rate $\eta$ to

$$\eta_b = \frac{\eta}{b},$$

and leaving other parts unchanged. Thus the generalization to batch-SGD is straightforward—simply replacing all $\eta$ with $\eta_b$.

### D.2 The SDE of $\theta$

In this section, we carefully derive the continuous dynamics of SGD for the parameterization of our rank-1 linear networks. We first discuss the balanced initialization condition.

**Balanced initialization.** Similar to the case for GD (Definition 2), we also assume the balanced initialization Eq. (26) across layers. Although the dynamics of SGD is different with that of GD, it still applies the gradient to update the parameters at every step that will maintain the balanced property thus the dynamics of SGD will also maintain the balanced property, i.e.,

$$\frac{\langle v_{k+1}(t), u_k(t)\rangle^2}{\|v_{k+1}(t)\|^2\|u_k(t)\|^2} = 1$$

and

$$\|v_{k+1}(t)\| = \|u_k(t)\| = \|v_1(t)\|$$

for $t > 0$ and $k \in \{1, \ldots, L-1\}$, during the training of rank-1 linear networks.

With the equations for $u_k$ and $v_{k+1}$, we now derive the SDE of $\theta$ summarized in the following lemma.

---

[2]More details of this modelling technique can be found in [14].

**Lemma 3.** *For an L-layer rank-1 linear network Eq. (2), if we assume balanced initialization, then the stochastic gradient flow of $\theta$ is*

$$d\theta = -\frac{2\xi^2}{n}H(\theta)X^T r dt + 2\xi^2\sqrt{\frac{\eta\mathcal{L}}{n}}H(\theta)X^T d\mathcal{W}_t$$
$$+ \frac{8\eta\mathcal{L}(L-1)}{n\|\theta\|^{\frac{2}{2L-1}}}\left[I + \frac{2L-3}{2}\frac{\theta\theta^T}{\|\theta\|^2}\right]X^T X\theta dt$$

*where $H(\theta) = I + 2(L-1)\theta\theta^T/\|\theta\|^2$.*

*Proof.* According to the Ito's Lemma, we have

$$d\theta = d(\xi v_1) = d\xi v_1 + \xi dv_1 + d\xi dv_1. \tag{76}$$

Thus to obtain the SDE of $\theta$, we need to analyze every term of the above equation. We first give $d\xi$.

**The form of $d\xi$.** Let $\omega_k = v_{k+1}^T u_k$ and $\psi_k = \|v_{k+1}\|^2 + \|u_k\|^2$, we first characterize the SDE of $\omega_k$. According to the Ito's calculus, we obtain that

$$d\omega_k = d(v_{k+1}^T u_k) = \underbrace{u_k^T dv_{k+1} + v_{k+1}^T du_k}_{\clubsuit} + \underbrace{du_k^T dv_{k+1}}_{\diamondsuit}, \tag{77}$$

where, by applying Eq. (73) and Eq. (74) and noting that $(d\mathcal{W}_t)^2 = dt$,

$$\clubsuit = -\frac{2}{n}v_1^T X^T r\rho_k\rho_{-k}\psi_k dt + 2\sqrt{\frac{\eta\mathcal{L}}{n}}\rho_k\rho_{-k}\psi_k v_1^T X^T d\mathcal{W}_t$$
$$\diamondsuit = \frac{4\eta\mathcal{L}}{n}(\rho_k\rho_{-k})^2\omega_k v_1^T X^T X v_1 dt.$$

Combining the above two terms gives us the SDE of $\omega_k$

$$d\omega_k = \left[-\frac{2}{n}v_1^T X^T r\rho_k\rho_{-k}\psi_k + \frac{4\eta(\rho_k\rho_{-k})^2\mathcal{L}}{n}\omega_k v_1^T X^T X v_1\right]dt + 2\sqrt{\frac{\eta\mathcal{L}}{n}}\rho_k\rho_{-k}\psi_k v_1^T X^T d\mathcal{W}_t.$$

Since $\xi = d(\prod_{k=1}^{L-1}\omega_k)$, its SDE can be done by repeatedly applying the Ito's Lemma and the SDE of $\omega_k$:

$$d\xi = d(\prod_{k=1}^{L-1}\omega_k)$$
$$= \underbrace{\sum_{k=1}^{L-1}\frac{\xi}{\omega_k}d\omega_k}_{\spadesuit} + \frac{1}{2}\sum_{k',k=1,k\neq k'}^{L-1}\underbrace{\frac{\xi}{\omega_k\omega_{k'}}d\omega_k d\omega_{k'}}_{\heartsuit}. \tag{78}$$

For convenience, we first define several helper notations:

$$\Phi_1 = \sum_{k=1}^{L-1}\phi_k = \sum_{k=1}^{L-1}\frac{\psi_k}{\omega_k^2},$$
$$\Phi_2 = \sum_{k=1}^{L-1}\frac{1}{\omega_k^2},$$
$$\Phi_3 = \frac{1}{2}\sum_{k,k'=1,k\neq k'}^{L-1}\frac{\psi_k\psi_{k'}}{\omega_k^2\omega_{k'}^2}.$$

Now plugging the form of $d\omega_k$ into ♠ gives us the first term of $d\xi$:

$$\spadesuit = -\frac{2}{n}v_1^T X^T r\xi^2 \left(\sum_{k=1}^{L-1}\frac{\psi_k}{\omega_k^2}\right)dt + \frac{4\eta\mathcal{L}\xi^3 v_1^T X^T X v_1}{n}\left(\sum_{k=1}^{L-1}\frac{1}{\omega_k^2}\right)dt$$

$$+ 2\sqrt{\frac{\eta\mathcal{L}}{n}}\xi^2\left(\sum_{k=1}^{L-1}\frac{\psi_k}{\omega_k^2}\right)(v_1^T X^T d\mathcal{W}_t)$$

$$= -\frac{2}{n}v_1^T X^T r\xi^2\Phi_1 dt + \frac{4\eta\mathcal{L}\xi^3 v_1^T X^T X v_1}{n}\Phi_2 dt + 2\sqrt{\frac{\eta\mathcal{L}}{n}}\xi^2\Phi_1\left(v_1^T X^T d\mathcal{W}_t\right), \quad (79)$$

and applying again $(d\mathcal{W}_t)^2 = dt$ and the form of $d\omega_k$ gives us each term of the second sum of $d\xi$:

$$\heartsuit = \frac{\xi}{\omega_k\omega_{k'}}\frac{4\eta\xi^2\mathcal{L}}{n\omega_k\omega_{k'}}\psi_k\psi_{k'}v_1^T X^T X v_1 dt.$$

Summing all $\heartsuit$ and ♠, we obtain the SDE of $\xi$:

$$d\xi = \left[-\frac{2}{n}v_1^T X^T r\xi^2\Phi_1 + \frac{4\eta\mathcal{L}\xi^3 v_1^T X^T X v_1}{n}\Phi_2 + \frac{4\eta\xi^3\mathcal{L}v_1^T X^T X v_1}{n}\Phi_3\right]dt$$

$$+ 2\sqrt{\frac{\eta\mathcal{L}}{n}}\xi^2\Phi_1\left(v_1^T X^T d\mathcal{W}_t\right)$$

$$= \left[-\frac{2}{n}v_1^T X^T r\xi^2\Phi_1 + \frac{4\eta\mathcal{L}\xi^3 v_1^T X^T X v_1}{n}(\Phi_2 + \Phi_3)\right]dt + 2\sqrt{\frac{\eta\mathcal{L}}{n}}\xi^2\Phi_1\left(v_1^T X^T d\mathcal{W}_t\right). \quad (80)$$

The SDE of $v_1$ is much simpler. To get this, we start with the SGD update equation for $v_1$:

$$v_1(t+1) = v_1(t) - \eta\frac{\partial\mathcal{L}}{\partial v_1} + \eta\left(\frac{\partial\mathcal{L}}{\partial v_1} - \frac{\partial\ell_{j_t}}{\partial v_1}\right)$$

$$= v_1(t) - \frac{2\eta}{n}\xi X^T r + 2\eta\left(\frac{1}{n}\sum_i \xi x_i r_i - r_{j_t}\xi x_{j_t}\right)$$

$$= v_1(t) - \frac{2\eta}{n}\xi X^T r + 2\eta\xi X^T Z_{j_t}, \quad (81)$$

which implies that the noise covariance in this case is

$$\Sigma(v_1(t)) = \frac{4\eta^2\xi^2\mathcal{L}}{n}X^T X.$$

Then using a similar approach as that of $u_k$, we get the SDE of $v_1$

$$dv_1 = -\frac{2}{n}\xi X^T r dt + 2\sqrt{\frac{\eta\mathcal{L}}{n}}\xi X^T d\mathcal{W}_t. \quad (82)$$

Now it is sufficient for us to derive the form of $d\theta$.

**The form of $d\theta$.** Combined with the SDE of $\xi$, we now have

$$d\theta = d(\xi v_1) = \underbrace{\xi dv_1 + v_1 d\xi}_{\clubsuit} + \underbrace{d\xi dv_1}_{\spadesuit}. \quad (83)$$

For the ♣ term, as we already have the form of $dv_1$ in Eq. (82) and $d\xi$ in Eq. (80), we simply plug them into ♣ and obtain that:

$$\clubsuit = -\frac{2}{n}\xi^2 X^T r dt + 2\sqrt{\frac{\eta\mathcal{L}}{n}}\xi^2\left(X^T d\mathcal{W}_t\right) + v_1\left[-\frac{2}{n}v_1^T X^T r\xi^2\Phi_1 + \frac{4\eta L\xi^3 v_1^T X^T X v_1}{n}(\Phi_2 + \Phi_3)\right]dt$$

$$+ 2\sqrt{\frac{\eta\mathcal{L}}{n}}\Phi_1\xi^2 v_1 v_1^T X^T d\mathcal{W}_t$$

$$= -\frac{2}{n}\left(\xi^2 I + \xi^2\Phi_1 v_1 v_1^T\right)X^T r dt + \frac{4\eta\mathcal{L}\xi^3 v_1 v_1^T X^T X v_1}{n}(\Phi_2 + \Phi_3)dt$$

$$+ 2\sqrt{\frac{\eta\mathcal{L}}{n}}\left(\xi^2 I + \xi^2\Phi_1 v_1 v_1^T\right)X^T d\mathcal{W}_t. \quad (84)$$

For the ♠ term, we only need to consider the $d\mathcal{W}_t$ terms of Eq. (82) and (80):

$$\spadesuit = 4\frac{\eta\mathcal{L}\xi^3}{n}\Phi_1 X^T X v_1 dt.$$

Combining the above two equations gives us the final SDE of $d\theta$:

$$d\theta = -\frac{2}{n}\left(\xi^2 I + \xi^2 \Phi_1 v_1 v_1^T\right) X^T r dt + \frac{4\eta\mathcal{L}}{n}\left[\xi^3 \Phi_1 I + \xi^3 (\Phi_2 + \Phi_3) v_1 v_1^T\right] X^T X v_1 dt$$
$$+ 2\sqrt{\frac{\eta\mathcal{L}}{n}}\left(\xi^2 I + \xi^2 \Phi_1 v_1 v_1^T\right) X^T d\mathcal{W}_t. \tag{85}$$

On the other hand, recall that our assumptions regarding the initial conditions of $f(x; u, v)$ in the Lemma (Definition 1) and following similar techniques as in the case for GD, we have that for any $t > 0$:

1. $\forall k : \|u_k(t)\|^2 = \|v_k(t)\|^2 = \|v_1(t)\|^2$

2. $\forall k : \omega_k^2 = \langle u_k(t), v_{k+1}(t)\rangle^2 = \|u_k(t)\|^4 = \|v_1(t)\|^4$.

Plugging these terms back to the definitions of $\Phi_1$, $\Phi_2$ and $\Phi_3$, we obtain that

$$\psi_k = 2\|v_1\|^2 \text{ and } \xi^2 = \|\theta\|^{\frac{4(L-1)}{2L-1}} \tag{86}$$

as in the case for GD and

$$\Phi_1 = \frac{2(L-1)}{\|v_1\|^2} = \frac{2(L-1)\xi^2}{\|\theta\|^2}, \tag{87}$$

$$\Phi_2 = \frac{L-1}{\|v_1\|^4} = \frac{\xi^4(L-1)}{\|\theta\|^4}, \tag{88}$$

$$\Phi_3 = \frac{2(L-1)(L-2)}{\|v_1\|^4} = \frac{2\xi^4(L-1)(L-2)}{\|\theta\|^4}, \tag{89}$$

where we use that $\|\theta\| = \|\xi v_1\| = |\xi|\|v_1\|$. Therefore, the final form of $d\theta$ is now

$$d\theta = -\frac{2\xi^2}{n}H(\theta)X^T r dt + 2\xi^2 \sqrt{\frac{\eta\mathcal{L}}{n}}H(\theta)X^T d\mathcal{W}_t$$
$$+ \frac{8\eta\mathcal{L}(L-1)}{n\|\theta\|^{\frac{2}{2L-1}}}\left[I + \frac{2L-3}{2}\frac{\theta\theta^T}{\|\theta\|^2}\right]X^T X\theta dt$$

where

$$H(\theta) = I + 2(L-1)\frac{\theta\theta^T}{\|\theta\|^2}.$$

$\square$

## D.3 Proof for Theorem 3

In this section, we determine the form of $V^{\mathcal{S}}(\theta)$ such that $\theta$ follows a stochastic mirror flow

$$d\partial_\theta V^{\mathcal{S}}(\theta, t) = -\frac{\partial\mathcal{L}}{\partial\theta}dt + 2\sqrt{\frac{\eta\mathcal{L}}{n}}(X^T d\mathcal{W}_t), \tag{90}$$

which then proves the claims of Theorem 3. For this purpose, we start with manipulating the SDE of $\theta$ derived in Lemma 3. Note that Eq. (85) can be written as

$$\left(\xi^2 I + \xi^2 \Phi_1 v_1 v_1^T\right)^{-1} d\theta - \frac{4\eta\mathcal{L}}{n}\mathcal{P}dt = -\frac{2}{n}X^T r dt + 2\sqrt{\frac{\eta\mathcal{L}}{n}}X^T d\mathcal{W}_t, \tag{91}$$

where

$$\mathcal{P} = \left(\xi^2 I + \xi^2 \Phi_1 v_1 v_1^T\right)^{-1}\left[\xi^3 \Phi_1 I + \xi^3 (\Phi_2 + \Phi_3)v_1 v_1^T\right]X^T X v_1.$$

To solve the inverse appeared in the above equation, we apply Lemma 2 and noting that

$$\mathrm{tr}\left(\xi^2 \Phi_1 v_1 v_1^T \xi^{-2} I\right) = \Phi_1 \|v_1\|^2,$$

then

$$
\begin{aligned}
\left(\xi^2 I + \xi^2 \Phi_1 v_1 v_1^T\right)^{-1} &= \frac{1}{\xi^2} I - \frac{1}{1 + \Phi_1 \|v_1\|^2} \frac{\xi^2 \Phi_1 v_1 v_1^T}{\xi^4} \\
&= \frac{1}{\xi^2}\left(I - \frac{\Phi_1 v_1 v_1^T}{1 + \Phi_1 \|v_1\|^2}\right) \\
&= \frac{1}{\xi^2}\left(I - \frac{1}{1 + \frac{\xi^2}{\Phi_1 \|\theta\|^2}} \frac{\theta\theta^T}{\|\theta\|^2}\right),
\end{aligned}
$$

where we use that $\theta = \xi v_1$ in the last equality, which enables us to simplify $\mathcal{P}$:

$$
\begin{aligned}
\mathcal{P} &= \left(\xi^2 I + \xi^2 \Phi_1 v_1 v_1^T\right)^{-1}\left[\xi^3 \Phi_1 I + \xi^3(\Phi_2 + \Phi_3)v_1 v_1^T\right] X^T X v_1 \\
&= \Phi_1\left(I - \frac{1}{1 + \frac{\xi^2}{\Phi_1 \|\theta\|^2}} \frac{\theta\theta^T}{\|\theta\|^2}\right)\left[I + \frac{\Phi_2 + \Phi_3}{\xi^2 \Phi_1} \theta\theta^T\right] X^T X \theta \\
&= \Phi_1\left[I - \left(\frac{1}{\|\theta\|^2 + \frac{\xi^2}{\Phi_1}} - \frac{\Phi_2 + \Phi_3}{\xi^2 \Phi_1} + \frac{\|\theta\|^2}{\|\theta\|^2 + \frac{\xi^2}{\Phi_1}} \frac{\Phi_2 + \Phi_3}{\xi^2 \Phi_1}\right)\theta\theta^T\right] X^T X \theta \\
&= \Phi_1\left[I - \frac{\Phi_1^2 - \Phi_2 - \Phi_3}{\Phi_1^2 \|\theta\|^2 + \xi^2 \Phi_1}\theta\theta^T\right] X^T X \theta.
\end{aligned}
$$

Thus Eq. (91) for the overall SDE of $\theta$ now becomes

$$
\frac{1}{\xi^2}\left(I - \frac{1}{1 + \frac{\xi^2}{\Phi_1 \|\theta\|^2}} \frac{\theta\theta^T}{\|\theta\|^2}\right) d\theta - \frac{4\eta\mathcal{L}}{n}\mathcal{P} dt = -\frac{\partial\mathcal{L}}{\partial\theta} dt + 2\sqrt{\frac{\eta\mathcal{L}}{n}}(X^T d\mathcal{W}_t). \tag{92}
$$

The balanced initialization gives us Eq. (87), (88), and (89), and recall that

$$\lambda_L = \frac{2(L-1)}{2L-1},$$

thus we can further rewrite

$$
\begin{aligned}
\frac{1}{\xi^2}\left(I - \frac{1}{1 + \frac{\xi^2}{\Phi_1 \|\theta\|^2}} \frac{\theta\theta^T}{\|\theta\|^2}\right) &= \frac{1}{\xi^2}\left(I - \frac{1}{1 + \frac{1}{2(L-1)}} \frac{\theta\theta^T}{\|\theta\|^2}\right) \\
&= \frac{1}{\xi^2}\left(I - \lambda_L \frac{\theta\theta^T}{\|\theta\|^2}\right) \tag{93}
\end{aligned}
$$

and

$$
\begin{aligned}
\mathcal{P} &= \frac{1}{\xi^2}\left(I - \lambda_L \frac{\theta\theta^T}{\|\theta\|^2}\right)\left[\xi^2 \Phi_1 I + (\Phi_2 + \Phi_3)\theta\theta^T\right] X^T X \theta \\
&= \frac{1}{\xi^2}\left[\xi^2 \Phi_1 I + \left((\Phi_2 + \Phi_3)(1 - \lambda_L) - \frac{\lambda_L \xi^2 \Phi_1}{\|\theta\|^2}\right)\theta\theta^T\right] X^T X \theta \\
&= \frac{1}{\xi^2}\left[\frac{2(L-1)\xi^4}{\|\theta\|^2}I + ((L-1+4(L-1)(L-2))(1-\lambda_L) - 2(L-1)\lambda_L)\frac{\xi^4}{\|\theta\|^4}\theta\theta^T\right] X^T X \theta \\
&= \frac{\xi^2}{\|\theta\|^2}\left[2(L-1)I - \frac{3(L-1)}{2L-1}\frac{\theta\theta^T}{\|\theta\|^2}\right] X^T X \theta \\
&= \frac{2(L-1)\xi^2}{\|\theta\|^2}\left(I - \frac{\theta\theta^T}{2\|\theta\|^2}\right) X^T X \theta. \tag{94}
\end{aligned}
$$

Thus, Eq. (91) now[3] can be rewritten as

$$\frac{1}{\xi^2}\left(I - \lambda_L \frac{\theta\theta^T}{\|\theta\|^2}\right)d\theta - \frac{8\eta\mathcal{L}}{n}\frac{(L-1)\xi^2}{\|\theta\|^2}\left(I - \frac{1}{2}\frac{\theta\theta^T}{\|\theta\|^2}\right)X^T X\theta dt$$

$$= -\frac{\partial\mathcal{L}}{\partial\theta}dt + 2\sqrt{\frac{\eta\mathcal{L}}{n}}(X^T d\mathcal{W}_t). \tag{95}$$

**Finding the form of $V^\mathcal{S}(\theta)$.** We now proceed to find the form of $V^\mathcal{S}(\theta)$. For convenience, we apply a time re-scaling technique such that $d\tau = |\xi|dt$ ($d\mathcal{W}_\tau = \sqrt{|\xi|}d\mathcal{W}_t$ according to [12]) to the above equation. For convenience, we still use $t$ to represent the time after re-scaling. Then the above equation becomes

$$\frac{1}{\|\theta\|^{\frac{2(L-1)}{2L-1}}}\left(I - \lambda_L \frac{\theta\theta^T}{\|\theta\|^2}\right)d\theta - \frac{8\eta\mathcal{L}}{n}\frac{(L-1)\xi^2}{\|\theta\|^2}\left(I - \frac{1}{2}\frac{\theta\theta^T}{\|\theta\|^2}\right)X^T X\theta dt$$

$$= -\frac{\partial\mathcal{L}}{\partial\theta}dt + 2\sqrt{\frac{\eta|\xi|\mathcal{L}}{n}}(X^T d\mathcal{W}_t). \tag{96}$$

Recall that $\theta(0) \in \mathbb{R}^d$ is the initialization of $\theta$, since $V^\mathcal{S}(\theta)$ for SGD should have similar form as that for GD, we borrow from the GD results and first define a constant vector $\gamma \in \mathbb{R}^d$

$$\gamma = -\|\theta(0)\|^{-\frac{2L-2}{2L-1}}\theta(0).$$

Suppose now that $V^\mathcal{S}$ has the following form:

$$V^\mathcal{S}(\theta, t) = \frac{2L-1}{2L}\|\theta\|^{\frac{2L}{2L-1}} + \gamma^T\theta + g(t)^T\theta \text{ for } g \in \mathbb{R}^d, \tag{97}$$

by similar techniques as in the case of GD, we obtain the first and second derivatives of $V^\mathcal{S}(\theta)$ w.r.t $\theta$:

$$\partial_\theta V^\mathcal{S}(\theta, t) = \|\theta\|^{-\frac{2(L-1)}{2L-1}}\theta + \gamma + g(t),$$

$$\partial_\theta^2 V^\mathcal{S}(\theta, t) = \frac{1}{\|\theta\|^{\frac{2(L-1)}{2L-1}}}\left[I - \lambda_L \frac{\theta\theta^T}{\|\theta\|^2}\right].$$

There exists a function of $\theta$, $G(\theta)$ whose exact form can be derived but not necessary for us, corresponding to our $V^\mathcal{S}(\theta, t)$ defined above, such that $d\partial_\theta V^\mathcal{S}(\theta, t)$ can be written as

$$d\partial_\theta V^\mathcal{S}(\theta) = \partial_\theta\left(\partial_\theta V^\mathcal{S}(\theta, t)\right)d\theta + \partial_t\left(\partial_\theta V^\mathcal{S}(\theta, t)\right)dt + \frac{\partial^2}{\partial\theta\partial\theta}\partial_\theta V^\mathcal{S}(\theta, t)(d\theta)^2$$

$$= \partial_\theta\left(\partial_\theta V^\mathcal{S}(\theta, t)\right)d\theta + \left[\partial_t\left(\partial_\theta V^\mathcal{S}(\theta, t)\right) + G(\theta)\frac{\partial^2}{\partial\theta\partial\theta}\partial_\theta V^\mathcal{S}(\theta, t)\right]dt$$

Now suppose that we can choose a particular $g(t)$, thus a particular $G(\theta)$, such that the following relation is satisfied:

$$\frac{\partial}{\partial t}\partial_\theta V^\mathcal{S}(\theta, t) + G(\theta)\frac{\partial^2}{\partial\theta\partial\theta}\partial_\theta V^\mathcal{S}(\theta, t)$$

$$= -\frac{8\eta\mathcal{L}}{n}\frac{(L-1)\xi^2}{\|\theta\|^2}\left(I - \frac{1}{2}\frac{\theta\theta^T}{\|\theta\|^2}\right)X^T X\theta, \tag{98}$$

then we can rewrite the SDE of $\partial_\theta V^\mathcal{S}(\theta, t)$ as follows:

$$d\partial_\theta V^\mathcal{S}(\theta) = \partial_\theta\left(\partial_\theta V^\mathcal{S}(\theta, t)\right)d\theta + \partial_t\left(\partial_\theta V^\mathcal{S}(\theta, t)\right)dt + G(\theta)\frac{\partial^2}{\partial\theta\partial\theta}\partial_\theta V^\mathcal{S}(\theta, t)dt$$

$$= \frac{1}{\|\theta\|^{\frac{2(L-1)}{2L-1}}}\left(I - \lambda_L \frac{\theta\theta^T}{\|\theta\|^2}\right)d\theta - \frac{8\eta\mathcal{L}}{n}\frac{(L-1)\xi^2}{\|\theta\|^2}\left(I - \frac{1}{2}\frac{\theta\theta^T}{\|\theta\|^2}\right)X^T X\theta dt \tag{99}$$

---

[3]Note that when $L = 1$ we do not have the second $dt$ term in the LHS of the equation, this term is brought by adding layers to the model.

which will give us the desired "mirror flow" equation:

$$d\partial_\theta V^{\mathcal{S}}(\theta) = -\nabla_\theta \mathcal{L} dt + 2\sqrt{\frac{\eta|\xi|\mathcal{L}}{n}} X^T d\mathcal{W}_t.$$

It is now suffice for us to find the particular $g(t)$ that makes Eq. (98) satisfied. For convenience, recall that for a vector $a$ we use $a_\mu$ to denote its $\mu$-th component, we define the following helper notations:

$$k = 2\xi^2 \sqrt{\frac{\eta\mathcal{L}}{n}} \in \mathbb{R}, \tag{100}$$

$$B = \left[ I + 2(L-1)\frac{\theta\theta^T}{\|\theta\|^2} \right] X^T \in \mathbb{R}^{d\times n}, \tag{101}$$

$$d\theta_\mu = O_\mu dt + \sum_i B_{\mu i} d\mathcal{W}_{t,i}, \tag{102}$$

$$D_\mu(\theta,t) = \left( \frac{\partial V^{\mathcal{S}}(\theta,t)}{\partial \theta} \right)_\mu = \|\theta\|^{-\lambda_L}\theta_\mu + \gamma_\mu + g_\mu(t), \quad D(\theta,t) \in \mathbb{R}^d, \tag{103}$$

$$\delta_{\rho\sigma} = 1 \text{ if } \rho = \sigma \text{ otherwise } \delta_{\rho\sigma} = 0, \tag{104}$$

where the exact form of $O$ can be obtained from Lemma 3 but is not necessary for us. In the following, we use $D$ to represent $D(\theta,t)$ for convenience. For our purpose of choosing the particular $g(t)$, $dD(\theta,t) = d\partial_\theta V^{\mathcal{S}}(\theta,t)$ should match the R.H.S of Eq. (99). To make this relation clear, according to the Ito's Lemma, we obtain that

$$dD = \partial_t D dt + \partial_\theta D d\theta + \underbrace{\frac{1}{2}\sum_{\rho,\sigma} \frac{\partial^2 D}{\partial\theta_\rho \partial\theta_\sigma} d\theta_\rho d\theta_\sigma}_{\spadesuit},$$

where the last term is crucial. From the SDE of $\theta$ (Lemma 3), we have that the $d\theta_\rho d\theta_\sigma$ appeared in $\spadesuit$ can be written as

$$d\theta_\rho d\theta_\sigma = k^2 \sum_{j,i} B_{j\sigma}B_{\rho i} d\mathcal{W}_{t,i} d\mathcal{W}_{t,j} = k^2 \sum_i B_{i\sigma}B_{\rho i} dt.$$

On the other hand, according to

$$\partial_\theta D = \partial_\theta^2 V^{\mathcal{S}}(\theta,t) = \frac{1}{\|\theta\|^{\lambda_L}}\left( I - \lambda_L \frac{\theta\theta^T}{\|\theta\|^2} \right),$$

we have that, with explicit subscripts of vectors,

$$
\begin{aligned}
\frac{\partial^2 D_\mu}{\partial\theta_\rho \partial\theta_\sigma} &= \frac{\partial}{\partial\theta_\sigma}\left[ \|\theta\|^{-\lambda_L}\left( \delta_{\mu\rho} - \lambda_L \frac{\theta_\rho\theta_\mu}{\|\theta\|^2} \right) \right] \\
&= -\lambda_L\|\theta\|^{-\lambda_L-2}\delta_{\mu\rho}\theta_\sigma - \lambda_L\frac{\partial}{\partial\theta_\sigma}\left( \|\theta\|^{-\lambda_L-2}\theta_\rho\theta_\mu \right) \\
&= -\lambda_L\|\theta\|^{-\lambda_L-2}\delta_{\mu\rho}\theta_\sigma \\
&\quad - \lambda_L\left[ -(\lambda_L+2)\|\theta\|^{-\lambda_L-4}\theta_\sigma\theta_\mu\theta_\rho + \|\theta\|^{-\lambda_L-2}(\delta_{\rho\sigma}\theta_\mu + \delta_{\mu\sigma}\theta_\rho) \right] \\
&= -\frac{\lambda_L}{\|\theta\|^{\lambda_L+2}}(\delta_{\mu\rho}\theta_\sigma + \delta_{\mu\sigma}\theta_\rho) - \frac{\lambda_L}{\|\theta\|^{\lambda_L+2}}\theta_\mu\left( \delta_{\rho\sigma} - (\lambda_L+2)\frac{\theta_\rho\theta_\sigma}{\|\theta\|^2} \right).
\end{aligned} \tag{105}
$$

Note that the above expression implies that $\frac{\partial^2 D}{\partial\theta\partial\theta}$ is in fact a rank-3 tensor. Combined with the expression of $d\theta_\rho d\theta_\sigma$ derived above, we have that

$$\spadesuit = -\frac{k^2\lambda_L}{2\|\theta\|^{\lambda_L+2}}\sum_{\rho,\sigma}\sum_i \left[ \underbrace{\delta_{\mu\rho}\theta_\sigma + \delta_{\mu\sigma}\theta_\rho}_{H^\rho_{\mu\sigma}} + \theta^\mu\underbrace{\left( \delta_{\rho\sigma} - (\lambda_L+2)\frac{\theta_\rho\theta_\sigma}{\|\theta\|^2} \right)}_{P_{\rho\sigma}} \right] B_{\sigma i}B_{i\rho} dt \tag{106}$$

where $H \in \mathbb{R}^{d\times d\times d}$ is a rank-3 tensor. To get the exact form of $\spadesuit$, we need to have the exact forms of

**1.** $\sum_{\rho,\sigma,i} H^\rho_{\mu\sigma} B_{\sigma i} B_{i\rho}$. Note that $HBB \in \mathbb{R}^d$ and there are two different terms, both of which will induce a same vector. In particular,

$$\sum_{\rho,\sigma,i} \delta_{\rho\mu}\theta_\sigma B_{\sigma i}B_{i\rho} = \sum_{\rho\sigma,i} B_{\mu i}B_{i\sigma}\theta_\sigma = (BB^T\theta)_\mu$$

$$\sum_{\rho,\sigma,i} \delta_{\mu\sigma}\theta_\rho B_{\sigma i}B_{i\rho} = ((\theta^T BB^T)^T)_\mu = (BB^T\theta)_\mu.$$

Thus we have

$$\sum_{\rho,\sigma,i} H^\rho_{\mu\sigma} B_{\sigma i}B_{i\rho} = 2(B^TB\theta)_\mu,$$

where

$$
\begin{aligned}
BB^T\theta &= \left(I + 2(L-1)\frac{\theta\theta^T}{\|\theta\|^2}\right) X^TX \left(I + 2(L-1)\frac{\theta\theta^T}{\|\theta\|^2}\right)\theta \\
&= \left(I + 2(L-1)\frac{\theta\theta^T}{\|\theta\|^2}\right) X^TX \left(\theta + 2(L-1)\theta\right) \\
&= (2L-1)\left[I + 2(L-1)\frac{\theta\theta^T}{\|\theta\|^2}\right] X^TX\theta. \quad\quad (107)
\end{aligned}
$$

**2.** $\sum_{\rho,\sigma,i} P^\sigma_\rho B^i_\sigma B^\rho_i$. Using the matrix notation, we can easily find that

$$\sum_{\rho,\sigma,i} P^\sigma_\rho B^i_\sigma B^\rho_i = \mathrm{tr}\left(B^TPB\right),$$

where, recall the definition of $B$ in Eq. (101),

$$
\begin{aligned}
B^TPB &= X\left(I + 2(L-1)\frac{\theta\theta^T}{\|\theta\|^2}\right)\left(I - (\lambda_L+2)\frac{\theta\theta^T}{\|\theta\|^2}\right)\left(I + 2(L-1)\frac{\theta\theta^T}{\|\theta\|^2}\right) X^T \\
&= X\left[I + (2(L-1) - (\lambda_L+2) - 2(L-1)(\lambda_L+2))\frac{\theta\theta^T}{\|\theta\|^2}\right] \\
&\quad \times \left[I + 2(L-1)\frac{\theta\theta^T}{\|\theta\|^2}\right] X^T \\
&= X\left[I - 2(2L-1)\frac{\theta\theta^T}{\|\theta\|^2}\right]\left[I + 2(L-1)\frac{\theta\theta^T}{\|\theta\|^2}\right] X^T \\
&= X\left[I - (8L^2 - 10L + 4)\frac{\theta\theta^T}{\|\theta\|^2}\right] X^T. \quad\quad (108)
\end{aligned}
$$

Taking the trace of the above equation gives us the second term of ♠:

$$\mathrm{tr}\left(B^TPB\right) = \mathrm{tr}\left(X^TX\right) - (8L^2 - 10L + 4)\frac{\theta^T}{\|\theta\|^2}X^TX\theta.$$

Now removing the $\mu$ subscript of the derived $\sum_{\rho,\sigma,i} H^\rho_{\sigma,\mu} B_{\sigma i}B_{i\rho}$ and recovering the matrix notation, we have the final form of ♠ by combining it with the result of $\mathrm{tr}\left(B^TPB\right)$

$$
\begin{aligned}
♠ &= -\frac{k^2\lambda_L}{2\|\theta\|^{\lambda_L+2}}\left[2(2L-1)\left(I + 2(L-1)\frac{\theta\theta^T}{\|\theta\|^2}\right)X^TX\theta\right] \\
&\quad -\frac{k^2\lambda_L}{2\|\theta\|^{\lambda_L+2}}\left[\theta\,\mathrm{tr}\left(X^TX\right) - (8L^2-10L+4)\frac{\theta\theta^T}{\|\theta\|^2}X^TX\theta\right] \\
&= \frac{k^2}{2}\left[-\frac{\lambda_L}{\|\theta\|^{\lambda_L+2}}\mathrm{tr}\left(X^TX\right)\theta - \frac{4(L-1)}{\|\theta\|^{\lambda_L+2}}\left(I - \frac{L}{2L-1}\frac{\theta\theta^T}{\|\theta^2\|}\right)X^TX\theta\right]dt. \quad\quad (109)
\end{aligned}
$$

Thus the SDE of $D$ now becomes

$$dD = \partial_\theta D\,d\theta$$
$$+ \left[\partial_t D - \frac{k^2\lambda_L}{2\|\theta\|^{\lambda_L+2}}\mathrm{tr}\left(X^TX\right)\theta - \frac{2k^2(L-1)}{\|\theta\|^{\lambda_L+2}}\left(I - \frac{L}{2L-1}\frac{\theta\theta^T}{\|\theta^2\|}\right)X^TX\theta\right]dt$$

To find the particular $g(t)$ such that Eq. (98) is satisfied, we need to require that

$$\partial_t D - \frac{k^2 \lambda_L}{2\|\theta\|^{\lambda_L+2}} \text{tr}\left(X^T X\right)\theta - \frac{2k^2(L-1)}{\|\theta\|^{\lambda_L+2}}\left(I - \frac{L}{2L-1}\frac{\theta\theta^T}{\|\theta^2\|}\right)X^T X\theta$$
$$= -\frac{8\eta\mathcal{L}}{n}\frac{(L-1)\xi^2}{\|\theta\|^2}\left(I - \frac{1}{2}\frac{\theta\theta^T}{\|\theta\|^2}\right)X^T X\theta.$$

Recall that $D = \|\theta\|^{-\lambda_L}\theta + \gamma + g(t)$, we have

$$\partial_t D = g'(t),$$

which, noting that $k^2 = 4\xi^4\eta\mathcal{L}/n$, further gives us that the following relation should be satisfied

$$g'(t) - \frac{2\xi^4\eta\mathcal{L}}{n\|\theta\|^{\lambda_L+2}}\left[\lambda_L \text{tr}\left(X^T X\right)\theta + 4(L-1)\left(I - \frac{L}{2L-1}\frac{\theta\theta^T}{\|\theta^2\|}\right)X^T X\theta\right]$$
$$= -\frac{8\eta\mathcal{L}}{n}\frac{(L-1)\xi^2}{\|\theta\|^2}\left(I - \frac{1}{2}\frac{\theta\theta^T}{\|\theta\|^2}\right)X^T X\theta.$$

As a result of this, we can, noting that $\xi^2 = \|\theta\|^{2\lambda_L}$ from $\|\theta\|^2 = \xi^2\|v_1\|^2$, give the required $g(t)$ that makes Eq. (98) satisfied by solving the following equation:

$$g'(t) = \frac{2\lambda_L\eta \text{tr}\left(X^T X\right)\mathcal{L}\|\theta\|^{\lambda_L-2}}{n}\theta$$
$$+ \frac{8\eta(L-1)\mathcal{L}\|\theta\|^{\lambda_L-2}}{n}\left[I - I - \left(\frac{L}{2L-1} - \frac{1}{2}\right)\right]X^T X\theta$$
$$= \frac{2\lambda_L\eta\mathcal{L}\|\theta\|^{\lambda_L-2}}{n}\text{tr}\left(\left(I - \frac{\theta\theta^T}{\|\theta\|^2}\right)X^T X\right)\theta. \tag{110}$$

Now let the orthogonal projection operator of $\theta$ be $P_\perp(\theta) = I - \frac{\theta\theta^T}{\|\theta\|^2}$, then we can solve $g(t)$ as

$$g(t) = \frac{2\lambda_L\eta}{n}\int_0^t \mathcal{L}(\theta)\|\theta\|^{\lambda_L-2}\theta\text{tr}\left(P_\perp(\theta)X^T X\right)ds. \tag{111}$$

With this particular $g(t)$, we can then give $V^{\mathcal{S}}(\theta, t)$

$$V^{\mathcal{S}}(\theta, t) = \frac{1}{\Omega_L}\|\theta\|^{\Omega_L} + \theta^T\left[\frac{2\lambda_L\eta}{n}\int_0^t \mathcal{L}(\theta)\|\theta\|^{\lambda_L-2}\theta\text{tr}\left(P_\perp(\theta)X^T X\right)ds - \frac{\theta(0)}{\|\theta(0)\|^{\lambda_L}}\right] \tag{112}$$

that satisfies the relation

$$d\partial_\theta V^{\mathcal{S}}(\theta) = -\nabla_\theta \mathcal{L}dt + 2\sqrt{\frac{\eta\mathcal{L}\xi}{n}}X^T d\mathcal{W}_t.$$

Moreover, a particular interesting case is that, as $L \to \infty$,

$$\lim_{L\to\infty} V(\theta, t) = \|\theta\| + \theta^T\left[\frac{2\eta}{n}\int_0^T \mathcal{L}(\theta)\left\|P_\perp(\theta)X^T\right\|_F^2\frac{\theta}{\|\theta\|}ds - \frac{\theta(0)}{\|\theta(0)\|}\right]. \tag{113}$$

