# OpenReview forum: "Implicit Bias of (Stochastic) Gradient Descent for Rank-1 Linear Neural Network"
_NeurIPS.cc/2023/Conference — NeurIPS 2023 poster_

### Official Review · Reviewer_zKfq · 2023-06-13

**Soundness:** 3 good
**Presentation:** 3 good
**Contribution:** 2 fair
**Rating:** 5
**Confidence:** 4

**Summary:**

The paper describes the implicit bias of GD and (a continuous approximation of) SGD in balanced rank-1 linear networks, which are linear networks where every second layer has only 1 neuron and there is only one output neuron. This bias takes the form of a potential equal to one term that scales with the parameter norm, a second term that describes the influence of the initialization and a third term that describes the effect of SGD and scales with the learning rate.

It is then argued that this model is a better approximation of general linear networks than previous work which describes the implicit bias of diagonal linear networks (showing a sparsity bias for small initializations that vanishes for large initializations).

**Strengths:**

There are few models where the implicit bias of GD/SGD can be described with an explicit formula, it is nice to have an explicit formula. The mathematical results appear to be all correct.

The result are discussed in details, though I do not agree with some of the main conclusions made from the results.

**Weaknesses:**

I strongly disagree with the general point that the result presented in this paper are a better description of the implicit bias of general linear networks, outside of the case where there is only one output, and I think that the one output case is not very interesting as can be seen from a simple symmetry argument.

Let me first describe why single output linear networks are not very interesting. In this setting the only thing that matters is the value of $\theta$ orthogonal to the data span. Linear networks are invariant under orthogonal transformations of the inputs, thus if we rotate the first layer weights around the span of $X$ the final learned $\theta$ will be rotated by the same rotation. Thus assuming a rotation invariant initialization of the weights (which is the case for Gaussian initialization and approximately true for other traditional initializations), we know that the distribution of the learned $\theta$ is invariant under rotation, the expected $\theta$ will thus minimize the $L_2$ norm, and the only interesting thing is the amount of noise. This paper answers this second question (the amount of noise and also how it depends on the initialization $\theta_0$), but randomness w.r.t. the initialization of the parameters is always detrimental (the expected risk equals the risk of the expected predictor plus the variance). There is thus only disadvantage to using a deep linear network instead of a traditional linear model ($L=1$).

I therefore think that linear networks are only interesting when there are multiple outputs, in which case they exhibit a low-rank bias (see citation [13] from the paper) which cannot be observed when there is a single output. And I would argue that diagonal linear networks are closer to these multiple outputs linear network than the model presented in this paper. Indeed they both feature a similar transition between a NTK regime without sparsity for large initializations, and for small initialization a dynamic that jumps from saddle to saddle [Jacot et al., Saddle-to-Saddle Dynamics in Deep Linear Networks: Small Initialization Training, Symmetry, and Sparsity], leading to some form of sparsity. Of course the resulting sparsities are different, with a sparsity of the entries of the learned vector for diagonal networks and a sparsity of rank for linear networks.

The authors motivate the study of rank-1 linear networks for the insight they give into models we care about: general linear networks or more importantly nonlinear networks. But there already exists (incomplete) description of the implicit bias of linear networks [13], of shallow nonlinear networks [Bach, Breaking the Curse of Dimensionality
with Convex Neural Networks; Abbé et al., The merged-staircase property: a necessary and nearly sufficient condition for SGD learning of sparse functions on two-layer neural networks], and even for deep nonlinear networks [Jacot, Implicit Bias of Large Depth Networks: a Notion of Rank for Nonlinear Functions]. If the authors think that their approach better describes these bias or could lead to better descriptions, they should motivate this at least a little bit.

Furthermore, the fact that in these settings the implicit bias cannot be expressed explicitely suggests that the models such as diagonal linear networks and rank-1 linear networks where such explicit formula exist are exceptions, not the rule. I personnaly think that the techniques used to derive such explicit formulas have little hope of extending to these more complex cases, and some alternative techniques (as cited above) have already shown much more potential.

To summarize I think that the results, while correct and precise, only describe the implicit bias of linear networks in a very specific and not particularly interesting case. And I do not think the results of this paper are very representative of the more interesting cases, since they fail to capture any form of sparsity, even though different forms of sparsity have been observed and seem to play avery important role in these more interesting settings.

**Questions:**

If I understand correctly Theorem 1 and 2 state the same thing for rank-1 linear networks and general single output linear networks, so only the SGD result (Theorem 3) is restricted rank-1 linear networks. Do you expect a similar direct generalization for the SGD case?

Since you assume balancedness anyways, the Theorem 1 of [Arora et al., On the Optimization of Deep Networks: Implicit Acceleration by Overparameterization] shows that the widths of the network does not matter for the dynamics of $\theta$ as long as they are larger than 1. Since rank-1 networks are linear networks where every second width is 1, this suggests that everything should extend directly to general single output linear networks. In that case, I think it would be clearer to state everything for single output linear networks, this way there is no need to introduce a whole new model of rank-1 linear networks and argue that they behave similarly.

**Limitations:**

The authors only cite [13] but do not compare the results, nor explain why they observe something very different (no sparsity). They should explain that the reason they cannot observe rank sparsity is because they only have 1 output. Instead some of the conclusions from the paper could be interpreted as saying that linear networks have no sparsity bias (when they emphasize that linear networks minimize the $L_2$ norm instead of the $L_1$ norm) which we know is not true when there are multiple outputs [13].

---

> ### Author Rebuttal · Authors · 2023-08-08
>
> We thank the reviewer for the valuable comments and interesting questions. Below we address your concerns.
> - - -
> ### Weaknesses Part
> 1. **Merit of studying single-output linear networks.** We acknowledge that single-output linear networks are a special type of linear networks in the setting with rotational invariant initialization. However, we argue that single-output networks are representative prototypes for comparing the implicit of S(GD) for different architectures with balanced initialization. All the previous works [3,23,19] (citations in the paper) on diagonal linear networks studied the single-output case, as our rank-1 network has done. One of main contributions in the paper is to show that even in single-output scenario, different architectures exhibit very different implicit bias, i.e. diagonal linear networks implicit minimize $\ell_1$-norm while rank-1  and standard linear networks minimize $\ell_2$ norm up to their depth.
> 2. **Proxy of linear networks.** Through our analysis Theorem 1 and 2, rank-1 and standard linear network show the same implicit bias with GD, while diagonal linear network does not. Thus, *in term of implicit bias*, rank-1 net is a better proxy of standard linear network with sufficient tractability for analysis. This is not to downgrade diagonal linear network, and instead the point is to show that different architectures could induce different implicit bias.
>
>    Through this proxy, **another major contribution of our work is the implicit bias of SGD on rank-1 net to shed light on standard linear and nonlinear nets**. We characterize the joint effect induced by the model architecture, initialization, and stochastic sampling noise (Theorem 3). This implicit bias of SGD for rank-1 net is tractable for theoretical analysis while standard linear net is not.  We expect that the results for rank-1 linear networks can also be generalized to standard linear and non-linear networks. Indeed, we empirically verify such generalization in Fig. 5(c), Fig. 5(d), and Fig. 6(b), where, for standard linear networks, the solutions selected by SGD is closer to the $\ell_2$-norm solution than that selected by GD, and SGD exhibits a lower test loss. Furthermore, we also empirically verify this phenomenon for non-linear networks in Fig. 6(c), where SGD solutions enjoy lower test losses than GD solutions for all the different initialization scales. These results are consistent with the rank-1 linear networks and imply that SGD is less sensitive to the initialization scales than GD.
> 3. **Motivation of studying rank-1 network:** Complete and rigorous characterization of implicit bias of neural network is still an open problem in the deep learning community. Typically there exists a tradeoff between analytical tractability and generality when conducting theoretical underpinning. We admit that diagonal and rank-1 linear nets are special cases due to their analytical tractability, but still can shed light on standard linear and nonlinear networks, as demonstrated empirically.
> 4. **Comparison to [13]:** First, [13] investigated matrix factorization, i.e. regression with multiple-outputs,  showed  the learned matrix has a low-rank bias. On the other hand, both of our model and diagonal linear networks studied the overparameterized single-output linear regression. Thus, the setting is quite different. Second, it still remains as an open question that whether it is appropriate to compare the similarity of sparsity induced by diagonal linear net and matrix factorization, since the setting is different.
>
>     Our investigation on the implicit bias of (S)GD is designed in a prudent manner that we studied rank-1 model in the exactly same setting as diagonal linear network has done to compare the difference of their implicit bias.
> - - -
> ### Questions Part
> Since the rank-1 linear network and standard linear network show the same implicit bias with GD, we expect a similar generalization for the SGD case. From the empirical aspect, our numerical experiments show similar phenomenons for standard linear networks as in the case for rank-1 linear networks, which supports such generalization. Please see Fig. 5(c), Fig. 5(d), and Fig. 6(b) for the empirical observations.
>
> From the theoretical aspect, the analysis of SGD for rank-1 linear network is tractable, while the precise analysis for standard linear network (general single-output linear network) is still an open problem, which is one of reasons why we introduce the rank-1 linear network.
> - - -
> ### Limitations Part
> Please see our response to the Weaknesses 4 for the comparison of our work and [13].

---

> > ### Comment · Reviewer_zKfq · 2023-08-10
> >
> > I obviously see that technically speaking a one output linear network is more similar to a multiple output linear network than a diagonal network. But qualitatively speaking the one output case fails to capture any sparsity effect (and this is not surprising nor interesting) that we know exists in the multi-output case. I think this is a big limitations that should be discussed clearly, so that readers are not given the impression that fullt-connected linear networks have no sparsity bias (they have a rank sparsity bias no entry sparsity bias).
> >
> > Also I want to repeat, I am pretty sure that balanced rank-1 linear networks exhibit the exact same dynamics as balanced linear networks with one outputs. For gradient flow, this is follows directly from Theorem 1 in [Arora et al., On the Optimization of Deep Networks: Implicit Acceleration by Overparameterization] since this theorem proves that the dynamics do not depend on the widths as long as they are larger than 1 (in single output network). This makes your observation that balanced rank-1 linear networks and fully-connected linear networks have the same implicit bias a simple corollary of Theorem 1. Furthermore I am confident that the independence of the dynamics on the widths result can generalized to (S)GD or stochastic gradient flow, thus making the notion of balanced rank-1 linear networks completely obsolete.

---

> > > ### Author Response · Authors · 2023-08-11
> > >
> > > Thank you so much for the insightful discussion.
> > >
> > > We agree with the reviewer that the single output linear network does not characterize entry sparsity effect. To make this point clearer, we will state in detail the difference between the single output case and the multi-output case in the revision to reveal their different implicit bias on the aspect of sparsity.
> > >
> > > For the similarity between rank-1 linear networks and standard ones, we agree with the reviewer that they exhibit similar dynamics for gradient flow. Indeed, besides the perspective of the reviewer that treating rank-1 linear networks as standard linear networks where every second width is 1, we also formulate in Proposition 1 to illustrate their similarity in the sense that rank-1 linear networks are standard linear networks (every second width needs not to be 1) with special initialization under gradient flow.
> > >
> > > Mostly importantly, we would like to emphasize that results for gradient flow is not the most significant part of our work and we illustrate the motivation of the notion of rank-1 linear network and significance of our results here. We notice that there still lack fruitful results in the current literature regarding the analysis of the implicit bias of SGD (or SGF) for standard linear networks, which certainly has to do with the difficulty of characterizing the nature of the stochastic sampling noise. In particular, deriving the dynamics of the overall parameter $\theta = w_L^T W_{k - 1}\cdots W_1$ for standard linear networks under gradient flow is tractable, while deriving that for SGF is much more complicated: the Brownian motion term of the SDE of $\theta$ is composed of all the other SDEs of weight matrices $W_k$ in an intricate manner. This is much more complicated than only considering the first-order ODE in the GF case. Therefore, our motivation for rank-1 linear network is that it is a model close to standard linear network (thus a proxy of it as formulated in Proposition 1) and tractable for rigorous theoretical analysis for SGD at the same time. In this sense, we would like to clarify that the most significant part of our work is the characterization of the SGD solutions of rank-1 linear networks and, more importantly, the corresponding implicit bias analysis such as the mitigation of initialization scale effect jointly induced by the model architecture and sampling noise. All these new findings of SGD for rank-1 model could shed light on understanding standard linear networks.
> > >
> > > Finally, on one hand, we agree with the reviewer that the independence of the dynamics on the width results (and perhaps Proposition 1) might be generalized to SGD, which has been verified by our numerical experiments from the perspective of implicit bias. On the other hand, in our view, the exact characterization for such generality is still an open problem and requires rigorous analysis.

---

### Official Review · Reviewer_hVR8 · 2023-06-30

**Soundness:** 4 excellent
**Presentation:** 4 excellent
**Contribution:** 3 good
**Rating:** 6
**Confidence:** 4

**Summary:**

The authors introduce a new simple model of neural networks: the *rank-1 linear neural network*. This model consists of $L$ layers of linear networks, each of the form $u_kv_k^\top$ (a rank one matrix parameterized by two vectors).
The paper motivates this model by shortcomings of other simplified models (such as diagonal linear networks) and by the natural low rank structure of the solution found by GD over linear networs.
Assuming convergence of the gradient flow (and stochastic gradient flow respectively) to an interpolator, the solution found is entirely described by an implicit regularization problem that depends on the initialization and the depth of the network (and on the noise and the trajectory of the iterates respectively). These results are then discussed to derive some insights on the role of noise, depth and initialization.

**Strengths:**

Studying simple models of neural networks is a very interesting question, that has been extensively studied in the previous couple of years. In this line of work, the present submission takes an orthogonal step and introduces a new model, which I find very interesting.
Then, the authors try to relate their results to the roles of important parameters of (stochastic) gradient flow: noise, depth and initialization.


The analysis is classical: deriving a (stochastic) mirror flow (with varying potentials) evolution for the model $\theta$, in order to directly have the potential that the model minimizes. This approach and the whole paper are rigorous, state their results in a clear and concise way, with clear notation, leading to a paper that is easy to follow. I think that the writing is here a strength of the paper.

**Weaknesses:**

1) I am not totally convinced by the motivation of rank-1 linear networks. This model is motivated by noting that linear NNs are biased towards low-rank structure (the conjecture is that they minimize some kind of weighted nuclear norm, for small initializations). But this is the case for small initializations, where they are in the so-called ‘‘rich regime''. Thus, does imposing rank 1 + arbitrary scale really model better real NNs and linear NNs and their implicit biases than diagonal linear nets ?
Then, I kind of disagree with the noted shortcomings of diagonal linear networks: DLNs make appear a transition between rich and kernel regimes (Woodworth et al. Colt2020): for small initializations, sparse vectors are promoted ($\ell^1$ regularization), while for large initializations, they are in an $\ell^2$ regularization, which amounts to the lazy training regime. These phenomenons (rich and lazy regimes) appear in linear networks, but also in more complex architectures.

2) Throughout all the papers (and in particular in the title!), gradient flow and stochastic gradient flow (the processes studied in the paper) are replaced by GD and SGD, which is misleading ! We indeed expect GD and SGD to be much more complicated than GF and SGF to study.
Related to this, I think that (i) GD and SGD should be replaced by GF and SGF when appropriate, and (ii) the authors should mention the recent work
*S) GD over Diagonal Linear Networks: Implicit Regularisation, Large Stepsizes and Edge of Stability*, Even et al. 2023, that studies GD and SGD (with non infinitesimal learning rates) on DLNs, when discussing the implicit bias of GD and SGD over DLNs in related works.
Indeed, this work derives the implicit bias for GD and SGD, and show that the nature of the two is totally different from GF in the rich regime.


**Questions:**

3) About the results of the implicit bias, I think the influence of the different parameters should be related to phenomenons that arise in more complex networks. Can the influence of scale be related to any rich or lazy regimes ? Or does imposing rank-1 matrices forbids this ?Also, for $L\to\infty$ in theorem 1, does this amount to aligning the solution to the initialization (for any scale) ?

4) Can the tools used by *S) GD over Diagonal Linear Networks: Implicit Regularisation, Large Stepsizes and Edge of Stability*, Even et al. 2023, to derive implicit biases with non-infinitesimal stepsizes in the rank-1 linear model ? If this does not seem too hard, I think this could be a good remark to write (just as a remark, I do not suggest the authors do the whole analysis agai of course).

Overall, I think the paper is well written. The maths and the results are easy to follow, and the ideas are novel and interesting. However, I have several concerns regarding the relevance of the proposed model. I think it is a bit overstated (see point 1. above). If the authors modify their paper in a revised version in order to include suggested comments, I think this paper should be accepted, for the originality of the proposed model. I don't think (I am quite sure in fact, but maybe the authors can change my mind about it) that this model is more relevant than DLNs as proxy for more complex architectures, so I think the model could be just motivated as a step towards the implicit bias of linear networks, and not as better proxy than DLNs.

**Limitations:**

No such limitations.

---

> ### Author Rebuttal · Authors · 2023-08-08
>
> We thank the reviewer for the valuable suggestions and appreciation of our work. Below we address your concerns.
> - - -
> ### Weaknesses Part
> 1. **On the motivation of rank-1 linear networks:** We would like to first clarify that we do not interpret the rank-1 linear network as an overall better proxy than diagonal linear networks (DLNs). Instead, we claim that the rank-1 linear network is a better proxy for *standard linear networks* from the perspective of implicit bias of gradient flow. This is because they share similar bias for both large and small initialization and rank-1 linear networks are in fact standard linear networks with special initialization (Proposition 1). These are *differences* between rank-1 linear networks and DLNs, *not shortcomings* of DLNs, which reveal that the implicit bias is affected by the difference between the architectures of rank-1 and diagonal linear networks. And we agree with the reviewer that the transition between rich and lazy regime of DLNs is meaningful and significant. We will make this statement clearer in the revision as suggested by the reviewer.
>
>     In this sense, we believe that the rank-1 linear network can be an alternative prototype model as DLNs for the theoretical understanding of deep learning models, e.g., it allows us to  characterize the joint effect induced by the architecture and sampling noise of SGF.
> 2. **Regarding the misleading usages of GD and SGD:**
>     - **On the suggestion (i).** Thanks for pointing this out. We agree with the reviewer that the using of GD and SGD, instead of GF and SGF, is misleading, and that GD and SGD might be more complicated than GF and SGF. We will fix this in the revision by suitably replacing GD and SGD with GF and SGF.
>     - **On the suggestion (ii).** Thanks for suggesting this nice related work, which reveals the difference between (S)GD and the infinitesimal step-size version, (S)GF. The observation that moderate step-size induces an equivalent initialization reduction effect (thus the solution is closer to the $\ell_1$-norm solution when compared to the infinitesimal step-size version) is highly enlightening. We will include the discussion of this related work in the revision.
> - - -
> ### Questions Part
> 3. **Rich or lazy regimes:** Imposing rank-1 matrices does not forbid the influence of scale of the initialization. For rank-1 linear networks with finite $L$, when the initialization is large, the lazy regime is also reached. As the initialization becomes smaller, we "escape" from the lazy regime and enter in a non-NTK regime. Below we present a detailed discussion.
>    - **(i) Large initialization.** We first discuss the large initialization case. According to Woodworth et al., COLT2020, for any $D$-homogeneous model ($D$ is a positive integer), the lazy regime (or NTK regime) is reached for large initialization. Since the rank-1 linear network is a homogeneous model, the NTK regime is reached for large initialization and the implicit bias is given by the RKHS norm predictor accordingly. In particular, let $x$ and $x'$ be two arbitrary inputs and $\theta(0)$ be the initialization, we can derive the NTK kernel $K(x, x')$ for the $L$-layer rank-1 linear networks:
>     $$
>         K(x, x') \propto \left< x, \left(I + 2 (L - 1)\frac{\theta(0)\theta^T(0)}{||\theta(0)||^2_2}\right)x'\right>,
>     $$
>     which is fixed during training for large initialization. Please see Fig. 2 for how increasing the initialization scale and the number of layer affect the shape of the corresponding mirror flow potential.
>    - **(ii) Small initialization.** For small initialization, according to Corollary 1.1, an $\ell_2$-norm minimization predictor which can not be captured by the NTK is returned. Therefore, as the initialization becomes smaller, we "escape" from the lazy regime.
>    - **(iii) Comparison to the rich regime.** Since the solution of rank-1 linear network is still captured by a kernel for small initialization regime, we may not claim that this regime is the same as the rich regime in DLNs. Instead, we should interpret it as a non-NTK regime.
>
>     Finally, $L \to \infty$ (for any finite scale of the initialization) in Theorem 1 has an effect of aligning the solution (if it is an interpolation solution) to the initialization. This is because, as $L \to \infty$, the corresponding mirror flow potential is composed of two parts, the $\ell_2$ norm plus a term that is minimized when the solution aligns with the initialization.
> 4. **On the Analysis of non-infinitesimal step size:** As motivated by Even et al., 2023, we need to derive the Stochastic Mirror Descent recursion with time varying potentials to generalize to the moderate step-size analysis. Since the proof technique of Even et al., 2023 does not seem to restricted to DLNs, we believe such generalization is possible.
> 5. **On the overall comment:** We thank the reviewer again for the valuable comments, and we will modify the paper accordingly as suggested by the comments in the revision.

---

> > ### Comment · Reviewer_hVR8 · 2023-08-17
> >
> > I thank the authors for their answers, clarifications, and suggested modifications. I keep my good opinion of this paper unchanged.

---

### Official Review · Reviewer_B6bN · 2023-07-02

**Soundness:** 3 good
**Presentation:** 4 excellent
**Contribution:** 3 good
**Rating:** 6
**Confidence:** 4

**Summary:**

The paper investigates the implicit biases of gradient descent (GD) and stochastic gradient descent (SGD) when training multilayer linear networks. It presents an analysis of specific potential functions that are minimized during the training process to achieve a perfect fit of the labels. Notably, the potential function's form is contingent on the depth of the network, providing insights into the impact of network depth on the implicit biases induced by GD/SGD.

**Strengths:**

-- The paper provides concrete results characterizing the actual norms that are being implicitly minimized by GD when training multilayer linear models. I think it is a valuable question to tackle as we lack an understanding of what are the actual implicit biases of GD and to understand what are the complexity measures that are being minimized by our optimizers.
-- The paper is well-written and discusses the literature quite well. It discussed the relevant papers that it is based on and its position against them.


**Weaknesses:**

-- The paper focuses on linear neural networks with a rank-1 constraint for each layer and assumes convergence to a perfect fit. While this specific setting is analyzed, it would be valuable to discuss the broader implications of the results. Considering that practical neural networks typically employ non-linear activations and do not necessarily have the rank-1 constraint, it would be interesting to explore if the findings can shed light on these more realistic scenarios.
-- The experimental details in the paper require clarification. It is unclear what specific settings were used, such as the optimization hyperparameters and the dataset employed (is it a real-world dataset or a synthetic one).
-- The paper is missing some relevant related work. For example, in https://arxiv.org/abs/2206.05794 they showed that when training a neural network with SGD and weight decay provably induces a low-rank bias in the weight matrices.


**Questions:**

Please see the weaknesses.

**Limitations:**

This paper has no negative societal impacts.

---

> ### Author Rebuttal · Authors · 2023-08-08
>
> Thank you for your valuable review and questions. Below we answer your questions.
> - - -
> ### Weaknesses Part
> 1. **On the broader implications of the results:** Since the rank-1 linear networks share similar implicit bias of gradient descent with standard linear networks, we expect the initialization reduction effect characterized by Theorem 3 for rank-1 linear networks can also be generalized to standard linear and nonlinear networks. Indeed, we empirically verify such generalization in Fig. 5(c), Fig. 5(d), and Fig. 6(b), where, for standard linear networks, the solutions selected by SGD is closer to the $\ell_2$-norm solution than that selected by GD, and SGD exhibits a lower test loss.
>
>     Furthermore, we also empirically verify this phenomenon for non-linear networks in Fig. 6(c), where SGD solutions enjoy lower test losses than GD solutions for all the different initialization scales. These results are consistent with that for rank-1 linear networks and imply that SGD is less sensitive to the initialization scales than GD.
> 2. **On the experimental details:** The dataset is a synthetic over-parameterizated regression dataset. Due to the lack of space, we defer all the experimental details to Appendix. Please refer to *Appendix A.1* of the supplementary materials for these details, where we specify all the dataset details and optimization hyper-parameters.
> 3. **Missing related works:** Thanks for suggesting this related work, which further supports our motivation of investigating rank-1 linear networks. We will include the discussion of it in the revision.

---

### Official Review · Reviewer_dEGz · 2023-07-08

**Soundness:** 4 excellent
**Presentation:** 4 excellent
**Contribution:** 3 good
**Rating:** 6
**Confidence:** 4

**Summary:**

This paper proposes the rank-1 linear network, a linear neural network where the even layers are restricted to row vectors and the odd layers are restricted to column vectors, as a new proxy of the deep neural networks.
In the first part, they gave two reasons why rank-1 linear networks are (said to be) close to linear networks:
(i) They analyze the deterministic gradient flow and find that the minimum L2-norm solution is preferred when the initialization scale is small, which is similar to linear neural networks and dissimilar to diagonal linear networks.
(ii) Also, they show that when a linear neural network is initialized to have a rank-k structure (k rows are non-zero in even layers and k-columns are non-zero in odd layers), this structure is kept during update with gradient flow, while diagonal structure is not kept.
In the second part, they analyzed the SDE update of rank-1 linear networks and give the explicit form of dynamics and the asymptotic solution.
The form includes a term depends on both stochasticity and depth, and implies that the effect of initialization gets smaller as the depth goes to infinity.


**Strengths:**

### The paper is well structured and provides convincing motivation for the setting, although the topic is a little niche.

(I think) the biggest result of this paper is the analysis of SGD solutions of rank-1 linear networks, and main critics would be about the validity of the setting and technical contributions.
That is, a rank-1 linear network is a special case of linear neural networks, and thus it is a "proxy of proxy" of practical deep neural networks, and the proofs are not so technically insightful because it somewhat heavily depends on the special initialization.
However, before considering the SGD dynamic, they spare a certain space on motivating the analysis of rank-1 linear networks by analyzing the GD dynamic and thoroughly comparing with linear neural networks than diagonal linear networks.
The discussion following each of theoretical results, which compares rank-1 linear networks with other architectures, is interesting, and this paper constructs an acceptable story on why this network architecture is important as a collection of theorems.

The "motivation" part mainly consists of two theorems.

- First, they derives the explicit form of the minimizer, which is attained as a result of deterministic gradient flow.
The form consists of the depth, L2-norm, and the inner-product with the initialization.
By changing the depth, different properties of the solution can be observed.
When the depth is finite, the solution minimizes a sum of L2-norm and a term reflecting initialization (among the zero-loss solutions).
As the network gets deeper, the effect of the latter gradually vanishes.
One of the important implications of this result is that the minimum L2-norm solution is preferred when the initialization scale is small, which is similar to 2-layer linear neural networks while diagonal linear networks are known to do so when the scale is large.

- Second, they show that rank-k property can be preserved in standard linear neural networks during optimization. They consider the setting when a linear neural network is initialized so that k rows are non-zero in even layers and k-columns are non-zero in odd layers, and is trained with deterministic gradient flow. They show that this structure is preserved during optimization, while if the linear network is initialized so that only diagonal elements are non-zero, the GF does not keep this structure.

In these senses, they argue that rank-1 restriction should be an acceptable proxy of linear neural networks, (compared to diagonal linear networks).

### It is an interesting result that the dependency on the scale of initialization can be mitigated by the depth and stochasticity.

As a result of the analysis of SDE dynamics of rank-1 linear networks, they obtain a transformation of the SDE and the explicit form of the asymptotic solution by letting the time go to infinity, under assuming the special initialization.
There exists a term that vanishes when either the depth is one or stochasticity is removed.
We can interpret this term (not by a rigorous mathematical proofs and with some leaps, though, ) as an implicit bias mitigating the effects from the initialization direction and making the solution closer to the minimum L2-norm solution compared to the GD under the same depth.

Overall, in both GD and SGD, they showed that letting the depth go to infinity decreases the effects of initialization, and in SGD, with some intuitive discussion, they showed that this effect is strengthened by stochasticity.
 Although this is only a possible "proxy of proxy" of DNNs in practice, but this paper at least poses an interesting question of whether this proposed benefit of depth/stochasticity holds in practical DNNs.

### Sufficient experiments as a theory paper.

Although I personally want to see more experimental results for non-linear DNNs, I think their main theorems are sufficiently verified with the experiments.

Overall, this paper is well-written. Although this is not a paradigm changing paper, I think that accepting this paper will attract ML theory community and encourage reconsideration of proxy-models.

**Weaknesses:**

### The balanced initialization is somewhat a strong assumption.

This paper assumes a special initialization of the network.
I know that this kind of assumptions are popular among the analysis of standard linear networks.
However, because the paper is proposing a new architecture as a proxy of standard linear networks and an alternative of diagonal linear network, and the main contribution is the new implicit bias jointly induced by stochasticity and depth, I think whether the implicit bias is unique to this initialization should be addressed.
I would like to know more experimental results about whether this initialization is necessary, and more theoretical discussion on this point.

### In Proposition 1., it seems that sparseness is more important than being rank-1. Thus I doubt that real-world NNs have this rank-1 structure.

It seems that even if rank(W_i) = 1 at initialization in Proposition 1., this does not guarantee that rank(W_i)=1 during optimization. Rather, having zero entries seems critical in proving the theorem. I think it is possible that matrices in NNs becomes rank(W_i)=1 during optimization, by using L2-reguralization and learning single-index models, for example. However, in most cases, it seems that real NNs cannot become this "rank-1 structure". Thus I am not sure whether this theorem is so meaningful.

**Questions:**

- When a small perturbation is added to the balanced initialization, then does the discrepancy between the balanced initialization and the perturbed one will grow during optimization? Are there theoretical or empirical observations?

- Can you derive the convergence rate guarantees?

**Limitations:**

The main limitations of this papers are that the architecture is a "proxy of proxy" of practical DNNs and that a special initialization is requited.
For the first, the author clearly remarks the limitation about extension to nonlinear settings, and I indeed think that this limitation is unavoidable.
For the latter. I hope the authors address the above question during the discussion period.

---

> ### Author Rebuttal · Authors · 2023-08-08
>
> Thank you for your detailed comments and valuable suggestions. Below we address your concerns.
> - - -
> ### Weaknesses Part
> 1. **On the balanced initialization:** We agree with the reviewer that the balanced initialization is special. To address this concern, we first present the empirical observations of the influence of adding a small perturbation to the balanced initialization, then theoretically discuss the main difficulty and effects of removing this assumption.
>
>
>    From the empirical aspect, we provide here a series of numerical experiments to show the influence of removing the balanced initialization. Specifically, we define
>     $$
>         \Delta = \frac{1}{2L - 1}\sum_{k = 1}^{L - 1} \frac{| ||v_{k + 1}||^2 - ||u_k||^2|}{||u_k||^2}
>     $$
>     as the scale of the perturbation to the balanced initialization. All the other experiment details are kept unchanged as in the Numerical Experiments section of our paper. We still observe similar phenomenons as in the case of the balanced initialization, e.g., SGD solutions are closer to the $\ell_2$-norm minimization solution compared to GD, when a small perturbation is added. Thus the implicit bias is not unique to the balanced initialization. Please see the figures in the attached PDF for the experimental results. In particular:
>
>       - We report $D(\theta(\infty), \theta_{\ell_2}^*)$ for both GD and SGD for different levels of perturbation $\Delta$ (denoted in the title of each figure) in the first four figures. In the first three figures, we use solid lines for the results of GD and dashed lines for SGD. For results under the balanced initialization, we use blue lines; for the results when a small perturbation is added to the balanced initialization, i.e., $\Delta \neq 0$, we use red lines. In the last figure of the first four figures, we fix the initialization scale and report $D((\theta(\infty), \theta_{\ell_2}^*)$ of both GD and SGD for different $\Delta$. It can be seen that, without the balanced initialization, GD and SGD still prefer $\ell_2$-norm minimization solution $\theta_{\ell_2}^*$ for small initialization, while the SGD solution is closer to $\theta_{\ell_2}^*$ due to its initialization reduction effect.
>
>     - We report $D(\theta(t), \theta_{\ell_2}^*)$ during optimization for both GD and SGD for different levels of perturbation $\Delta$ and the same scale of initialization ($||\theta(0)|| = 0.8696$) in the last figure. This figure further clearly reveals that there are still similar phenomenons when $\Delta\neq 0$ as in the case when the initialization is balanced.
>
>    From the theoretical aspect, balanced initialization enables us to derive the exact dynamics of the overall parameter $\theta$, which is necessary to precisely characterize the implicit bias of GD/SGD. In particular,
>     we need to solve a polynomial equation of the first layer weight $||v_1||$ of the form
>     $$
>         ||v_1||^2\left[\prod_k^{L - 1}(||v_1||^2 + \delta_k)(||v_1||^2 + \gamma_k) \right]- ||\theta||^2 = 0,
>     $$
>     where $\delta_k = ||v_{k + 1}(0)||^2 - ||v_1(0)||^2$ and $\gamma_k = ||u_{k}(0)||^2 - ||v_1(0)||^2$ are constants that measure the imbalance of the initialization. The main difficulty of removing the balanced initialization lies in that, according to the theory of polynomial equations, the above equation does not admit explicit solutions for general $L$ when $\delta_k\neq 0 $ and $\gamma_k \neq 0$ (for $L = 2$ we can explicitly solve the above equation and remove the balanced initialization assumption). Thus it is difficult to discuss arbitrary initialization without the balanced initialization assumption.
>
>     We may only asymptotically solve the above equation when $\delta_k$ and $\gamma_k$ are small perturbations, in the sense that $||v_1|| = ||\theta||^{1 / (2L - 1)} + \epsilon g(||\theta||, L)$ where $\epsilon$ is a small constant determined by $\delta_k$ and $\gamma_k$ and $g$ is a function of $||\theta||$ and $L$. In this way, the effect of removing the balanced initialization is that the induced mirror flow potential should be composed of two parts: the original potential presented in the paper and a perturbation due to the imbalance of the initialization to it. This implies that the $\ell_2$-norm solution is still returned for small initialization. On the other hand, the case for SGD is much more complicated: the Brownian motion term of the corresponding SDE will also be affected by the imbalance of the initialization, which in turn induces a much more complex time varying mirror flow potential. We believe that the exact theoretical characterization of the implicit bias of SGD without the balanced initialization is a valuable future direction.
>
>     Thanks again for the question on the balanced initialization, and we will include the above discussion in the revision.
> 2. **On Proposition 1:** We agree with the reviewer that only assuming rank$(W_i)$=1 does not guarantee rank$(W_i)$=1 during training, and that having zero entries indeed is critical in proving the theorem. Proposition 1 aims to show that rank-1 linear networks can be  cast as standard linear networks with special initialization, and that diagonal linear networks should be treated as a different architecture from standard linear network.
> - - -
> ### Questions Part
> 1. **On the small perturbation to the balanced initialization:** When a small perturbation is added to the balanced initialization, similar phenomenons still exist as in the case of balanced initialization. Please see our above response to Weakness 1 and the attached empirical results.
> 2. **On the convergence rates:** Since the corresponding optimization problem is a non-convex one and that the over-parameterization equivalently adds momentum to the dynamics of $\theta$, we believe that the convergence rate needs a thorough and fine analysis, which will become more complex when considering SGD. Since our focus is on the implicit bias aspect, we will leave such analysis for future works.

---

> > ### Comment · Reviewer_dEGz · 2023-08-21
> >
> > Dear authors,
> >
> > Thanks for the clarification. My concerns are appropriately addressed, and I consider the paper technically solid. I am therefore holding my score.
> >
> > Best regards,

---

### Author Rebuttal · Authors · 2023-08-08

We present the figures for the supplementary experiments in the attached PDF.

---

### Decision · Program_Chairs · 2023-09-21

**Decision:**

Accept (poster)

**Comment:**

This paper proposes to consider a theoretical model *rank-1 linear neural network* and analyzed the implicit bias of GD and SGD when the network converges to the interpolating solution. It is shown that GD and SGD minimize corresponding potential functions and the proposed frame-work is more consistent result to the standard linear network than well known simpler models such as diagonal linear network.
The idea is interesting and the theoretical founding of this paper sheds new light to the community. The numerical experiments well demonstrate the effectiveness of the approach.
In summary, this is a good paper. I would like to recommend acceptance.